# Dual-site segmentally synergistic catalysis mechanism: boosting CoFeS$_x$ nanocluster for sustainable water oxidation

Siran Xu[1,5], Sihua Feng[2,5], Yue Yu[1], Dongping Xue[1], Mengli Liu[1], Chao Wang[2], Kaiyue Zhao[3], Bingjun Xu[3] & Jia-Nan Zhang[1,4] ✉

Efficient oxygen evolution reaction electrocatalysts are essential for sustainable clean energy conversion. However, catalytic materials followed the conventional adsorbate evolution mechanism (AEM) with the inherent scaling relationship between key oxygen intermediates *OOH and *OH, or the lattice-oxygen-mediated mechanism (LOM) with the possible lattice oxygen migration and structural reconstruction, which are not favorable to the balance between high activity and stability. Herein, we propose an unconventional Co-Fe dual-site segmentally synergistic mechanism (DSSM) for single-domain ferromagnetic catalyst CoFeS$_x$ nanoclusters on carbon nanotubes (CNT) (CFS-ACs/CNT), which can effectively break the scaling relationship without sacrificing stability. Co$^{3+}$ (L.S, $t_{2g}^6 e_g^0$) supplies the strongest OH* adsorption energy, while Fe$^{3+}$ (M.S, $t_{2g}^4 e_g^1$) exposes strong O* adsorption. These dual-sites synergistically produce of Co-O-O-Fe intermediates, thereby accelerating the release of triplet-state oxygen ( ↑ O = O ↑ ). As predicted, the prepared CFS-ACs/CNT catalyst exhibits less overpotential than that of commercial IrO$_2$, as well as approximately 633 h of stability without significant potential loss.

The development of excellent electrocatalysts with high activity and long-term durability remains challenging due to destruction of the catalyst surface by the strong oxidizing environment during the oxygen evolution reaction (OER) process and its complex multi-step proton-coupling reaction[1–3]. The guiding principles for enhancing electrochemical performance are mainly based on the following strategies: (1) According to the adsorption evolution mechanism (AEM), OER activity is mainly influenced by the adsorption energetics of the oxygen intermediates (O*, OH* and OOH*) according to the Sabatier principle[4]. However, there is an inherent linear scaling relation (LSR) between the adsorption energies of reaction intermediates *OOH and *OH, thus rendering a minimum theoretical overpotential as high as ~0.4 eV even for the best possible material. (2) Compared with the conventional AEM, the lattice-oxygen-mediated mechanism (LOM)

overcomes the LSR constraint by triggering O-O coupling, which can decrease the energy barrier to the OER[5–7]. Nevertheless, in this mechanism, lattice oxygen is often required to be involved in O-O coupling, and accelerating the release of triplet-state oxygen. Additionally, the many oxygen vacancies lead to surface migration of the lattice oxygen, severe structural reconstruction and insufficient stability[8]. Therefore, catalysts based on LOM are not considered competitive with those based on AEM in terms of stability and feasibility for practical applications, despite their higher catalytic efficiency[9].

Alternatively, the oxide path mechanism (OPM) for homogeneous catalysts is more worthy of further OER performance improvements, since it allows direct O-O radical coupling without oxygen vacancy defects or additional reaction intermediates (*OOH). The O-O coupling

[1]Key Laboratory of Advanced Energy Catalytic and Functional Materials Preparation, College of Materials Science and Engineering, Zhengzhou University, Zhengzhou 450001, China. [2]National Synchrotron Radiation Laboratory, University of Science and Technology of China, Hefei 230026 Anhui, China. [3]College of Chemistry and Molecular Engineering, Peking University, Beijing, China. [4]State Key Laboratory of Coking Coal Resources Green Exploitation, Zhengzhou 450001, China. [5]These authors contributed equally: Siran Xu and Sihua Feng. ✉e-mail: zjn@zzu.edu.cn

mechanism require the presence of only O* and OH* intermediates during the OER process, which includes two types: (1) The O-O coupling generated by the lattice oxygen of the intrinsic catalyst and the solvent oxygen together (LOM); (2) The O-O coupling supplied by the solvent only (OPM). However, the LOM mechanism has complex trigger conditions, such as activation of lattice oxygen: The energy of the O $2p$ orbital must be moved up near the Fermi level to enable overlap with the central $d$-orbital of the metal (i.e., enhancing the covalency of the M−O bond), thus spanning the redox energy of the lattice oxygen[10,11]. Meanwhile, it is necessary to construct numerous holes to destabilize lattice oxygen atoms and enable the release of $O_2$. Compared with LOM, OPM process, regarded as an ideal pathway, allows a direct O-O coupling. In this process, $H_2O/OH^-$ is dissociated through the synergistic action of active metal sites, triggering M−O radical coupling and driving O-O coupling. However, this mechanism requires symmetric bimetallic positions with appropriate atomic distances to allow O-O radical coupling with low energy potential barriers[12,13]. However, OPM mechanism is less reported in alkaline environments, so exploring catalysts with appropriate polymetallic sites following the OPM mechanism in alkaline environments remains challenging. However, OPM mechanism is less reported in alkaline environments, so exploring catalysts with appropriate polymetallic sites following the OPM mechanism in alkaline environments remains challenging.

With the development of excellent catalysts from abundant $3d$ metals, external magnetic field disturbances affected the spin-catalyzed reaction kinetics and internal intrinsic spin diversity, allowing reaction pathways from spin-forbidden species (*OH, *OOH and *O to $O_2$) to spin-allowed species and accelerating the release of spin-parallel arrangement of triplet-state oxygen (↑O = O↑)[14–16]. Therefore, the construction of ferromagnetic catalysts is a prerequisite for achieving fast spin-catalyzed reaction kinetics. Among OER reaction, Co-Fe as typical ferromagnetic materials, show a leading edge with high OER activity in alkaline environments[17,18]. For catalytic reactions involving proton-coupled electron transfer (such as OER), the high surface energies of heterogeneous dual-atom catalysts and the low atomic utilization of crystalline bulk catalysts make it difficult to balance OER activity and stability. It is reasonable to infer that if the atomic size of the Co-Fe-based ferromagnetic catalyst was controlled to a single-domain region ( < 8 nm even clusters), spin-polarized OER would occur in the absence of a magnetic field[19]. Therefore, accelerating the release of parallel spin alignment triplet state oxygen by built-in electric field-induced localized ferromagnetic coupling is an effective pathway for superior OER catalysts[20]. Regarding this, the key intermediates (M-O*) followed by ferromagnetic coupling for a comprehensive understanding of the O-O direct coupling under the OPM mechanism provides a new perspective, and might be regarded as ultimately solving the primary stability and activity bottleneck of catalysts based on LOM or AEM.

In order to analyze the O-O coupling mechanism that can overcome the LSR and balance the high activity and strong stability, we chose ferromagnetic Co-Fe dual-site $CoFeS_x$ clusters supported on carbon nanotube CNT material as a platform to catalyze OER and elucidate the intrinsic relationship between the OER activity and the spin state of each metal site. Meanwhile, we suspected that focusing on the preferential adsorption of key oxygen intermediates on metals offers a promising research perspective for deepening understanding direct O-O coupling. The evolution of Fe(II) and Fe(III) spin states during the OER process is also considered and discussed.

## Results

### Rational design and synthesis
The $CoFeS_x$ nanoclusters supported on nanotubes (CFS-ACs/CNT) were synthesized via a facile adsorption reduction-hydrothermal method. First, amorphous $CoFeO_x$ nanosheets were prepared and supported onto pre-cleaned CNTs (CFO-p/CNT) via a $NaBH_4$ reduction

process[17], and strong interactions between metal precursors and rich oxygen functionalities on CNTs. Subsequently, $CoFeS_x$ nanoclusters were fabricated and maintained through the operable desired acidic environment generated by the decomposition of thioacetamide[20,21] during the hydrothermal process (Fig. 1a).

The chemical composition, morphology and chemical state of the final product were examined by X-ray powder diffraction (XRD) and transmission electron microscopy (TEM). The XRD patterns of CFS-ACs/CNT and CFO-p/CNT (Fig. S1) showed no obvious diffraction peaks belonging to Co(Fe)-based hybrids, which implied an amorphous structure because XRD is only sensitive to the crystalline materials[22]. Long-range disordered amorphous materials with a few atom exhibiting short-range order atoms tend to be endowed with more active sites. A broader range of chemical compositions, and greater structural flexibility than those of crystalline materials, and this leads to superior electrocatalytic activity[18]. Typically, TEM images of pure $CoFeO_x$ show the homogeneous nanosheets morphology (Fig. S2), and distinct CNTs with diameters of 10–20 nm are also observed in Fig. S3 with clear and complete lattice stripes. After $NaBH_4$ reduction, the TEM images (Fig. S4) showed that CFO-p/CNT retained the one-dimensional morphology of the CNTs, on which small $CoFeO_x$ nanosheets were uniformly decorated. Compared with the TEM image of the pure CNTs, the lattice stripes of the CNTs in the CFO-p/CNT catalyst show obvious distortions and fracture with due to embedding of the $CoFeO_x$ nanosheets (shown by yellow circles), which confirmed the successful introduction of $CoFeO_x$ species. With the introduction of sulfur source, the $CoFeO_x$ nanosheets were broken up and reconstituted into smaller CFS-ACs on the surfaces of the CNTs (Fig. 1b, c). In addition, atomic resolution high-angle annular dark-field scanning TEM (HAADF-STEM) was performed to investigate the distribution of CFS-ACs at the atomic scale. As shown in Fig. 1d, numerous well-dispersed CFS-ACs (marked by red circles) were present on CNTs support. The elemental maps further exhibited a uniform distributions of different atoms (Fe, Co, C, S) into CFS-ACs/CNT (Fig. 1e). The Co (17.63 wt%) and Fe (19.37 wt%) contents of CFS-ACs/CNT were determined with inductively coupled plasma-optical emission spectroscopy (ICP-OES), which was used as a standard for subsequent studies.

The X-ray photoelectron spectroscopy (XPS) was provided to further validate the coordination environment and chemical states of the all-prepared samples (Figs. S5–S9). In Fig. S5a, the Co $2p$ XPS spectra of CFO-p/CNT and CFS-ACs/CNT showed two major Co $2p_{3/2}$ and Co $2p_{1/2}$ spin-orbit splitting. The Co $2p$ peaks for CFS-ACs/CNT located at 779.9 (782.4 eV) and 794.9 eV (798.0 eV) were assigned to $Co^{3+}$ and $Co^{2+}$ respectively. Meanwhile, compared with the precursor, Fe/Co species of CFS-ACs/CNT exhibited positive peak shifts, suggesting a higher valence state. For the well-dispersed product, the presence of $Co^{3+}$ and the peripheral electron transfer of Co and Fe due to the introduction of S, further enhanced the OER activity[23,24]. After sulfidation, the metal oxidation state changed, while the whole O $1s$ spectrum was positively shifted (Fig. S6). The S $2p$ XPS spectrum (Fig. S7a) contained three peaks: Fe(Co)-S or C-S-C coordination at 162.5 eV, oxidized state S-O at 168.98 eV (S $2p_{3/2}$) and 170.24 eV (S $2p_{1/2}$)[25]. As shown in Fig. S7b, the peaks at 322, 340 and 375 $cm^{-1}$ were attributed to $T_g$, $E_g$ and $A_g$ modes of Fe-S and Co(Fe)-S bonds species, respectively, compared with the CFO-p/CNT[26]. Fourier Transform Infrared Spectrometer (FT-IR) spectrum of the CFS-ACs/CNT catalyst (Fig. S8) exposed a distinct stretching vibration of C-S bond. The presence of C-S-C further verified the introduction of S atoms into the CNTs. Meanwhile, the feature peaks of C $1s$ (Fig. S9) were shifted compared to those of the pure CNTs, which suggested that the strong metal-support interaction arising from the introduction of sulfur atoms led to the superior structural stability, higher conductivity and faster electron transfer during the OER. Raman spectroscopy was used to examine the intensity ratios ($I_D/I_G$) of the D-band to the G-band in the samples, indicating the surface defects in the carbon material (Fig. S10). The results showed that the $I_D/I_G$ ratio of CFS-ACs/CNT was

 

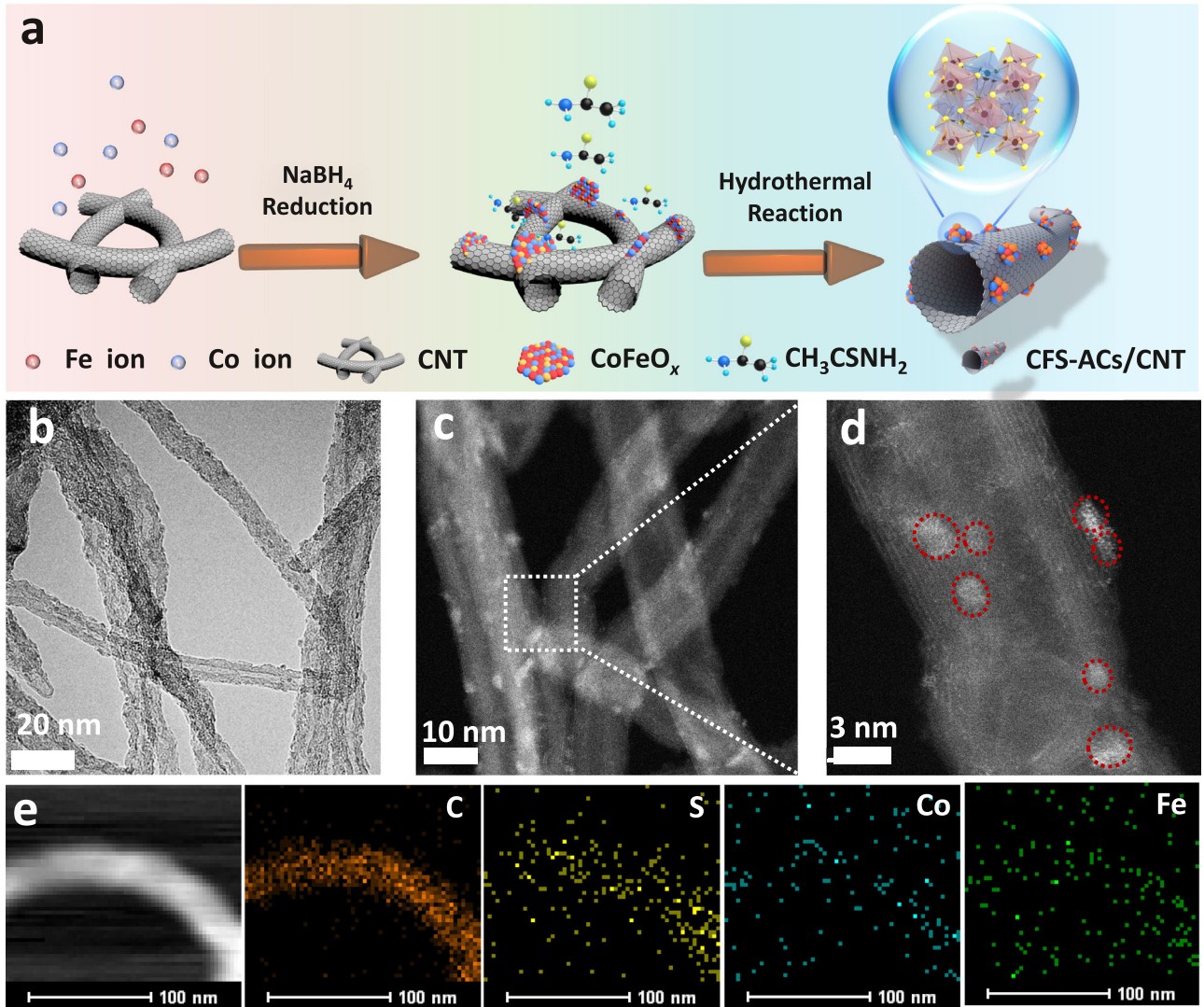

**Fig. 1 | Morphology and structures characterizations of CFS-ACs/CNT.**
**a** Synthesis and morphological characterizations of CFS-ACs/CNT. **b**, **c** Transmission electron microscopy (TEM) images. **d** Aberration corrected high angle annular dark-field (HAADF)-scanning transmission electron microscopy (STEM) images with zoom-in image showing a CoFeS$_x$ cluster (red circle), and **e** energy dispersive X-ray spectroscopy (EDS) mappings of individual elements (C, S, Fe, and Co).

smaller than that of CFO-p/CNT. Fewer levels of defects and the reduced disorder of the carbon materials suggested high site occupancy and well-dispersed metal active sites.

**Electron and spin state of the catalyst**

The X-ray absorption near-edge structure (XANES) of the Fe species in both CFS-ACs/CNT and FeS$_x$/CNT (Fig. 2a) showed the valence states FePc (+2) and Fe$_2$O$_3$ (+3). Meanwhile, CFS-ACs/CNT catalyst exhibited a slight shift to a lower photon energy, which indicated a chemical valence of approximately +2 in that sample, which was consistent with the XPS analysis. Similarly, the distinct pre-edge peak at ~7713.2 eV of Co K-edge XANES of CFS-ACs/CNT was assigned to the Co-O sample (Fig. S11a), while the absorption edge for the CFS-ACs/CNT catalyst was located between Co foil and Co-O, close to that of the phthalocyanine (CoPc)[27]. As depicted in Fig. 2b, the extended X-ray absorption fine structure (EXAFS) spectrum of CFO-p/CNT exposed a distinct predominant peak at 1.5 Å in $R$ space, which coincided with the Fe-O coordination of Fe$_2$O$_3$. Meanwhile, a dominant peak in the spectrum of CFS-ACs/CNT and FeS$_x$/CNT loaded at 1.68 Å was observed, which was situated between Fe-O and Fe-S (1.8 Å)[25,28]. Correspondingly, the highest Fe-Fe peak at 2.16 Å of Fe foil was absent for CFS-ACs/CNT.

Similarly, Co K-edge EXAFS spectrum of CFS-ACs/CNT (Fig. S11b) showed a negative shift with a predominant peak at 1.78 Å, which was located between Co-O (1.72 Å) and Co-S (1.83 Å)[29,30]. Moreover, the introduction of dual sites further aggravated the shortening of the M-S bond compared with the single metal site catalyst, which satisfied the trigger condition of the direct O-O coupling mechanism to a certain extent. As shown in Fig. S12, quantitative FT-EXAFS fitting was used to determine the structural parameters for the Fe(Co) local chemical environment. Co-S and Fe-S coordination numbers were 5.5 and 5.3 (Table S1), respectively, which implied the similar chemical environments of Co and Fe. For further verifying the structure, Fe K-edge XANES spectrum calculated based on CoFeS$_4$ structure was obtained, which coincided well with the Fe K-edge XANES spectra of CFS-ACs/CNT (Fig. S13). Furthermore, the wavelet transform EXAFS analysis (Fig. 2c and Fig. S14) of the Fe K-edge showed an intensity characteristic peak at 1.70 Å$^{-1}$, which was attributed to Fe-S coordination.

The unpaired electron in the Fe atomic orbitals in CFS-ACs/CNT and FeS$_x$/CNT samples were further identified with electron paramagnetic resonance (EPR) spectra. As shown in Fig. S15, the EPR peak signal was more pronounced after the introduction of Co[31], which

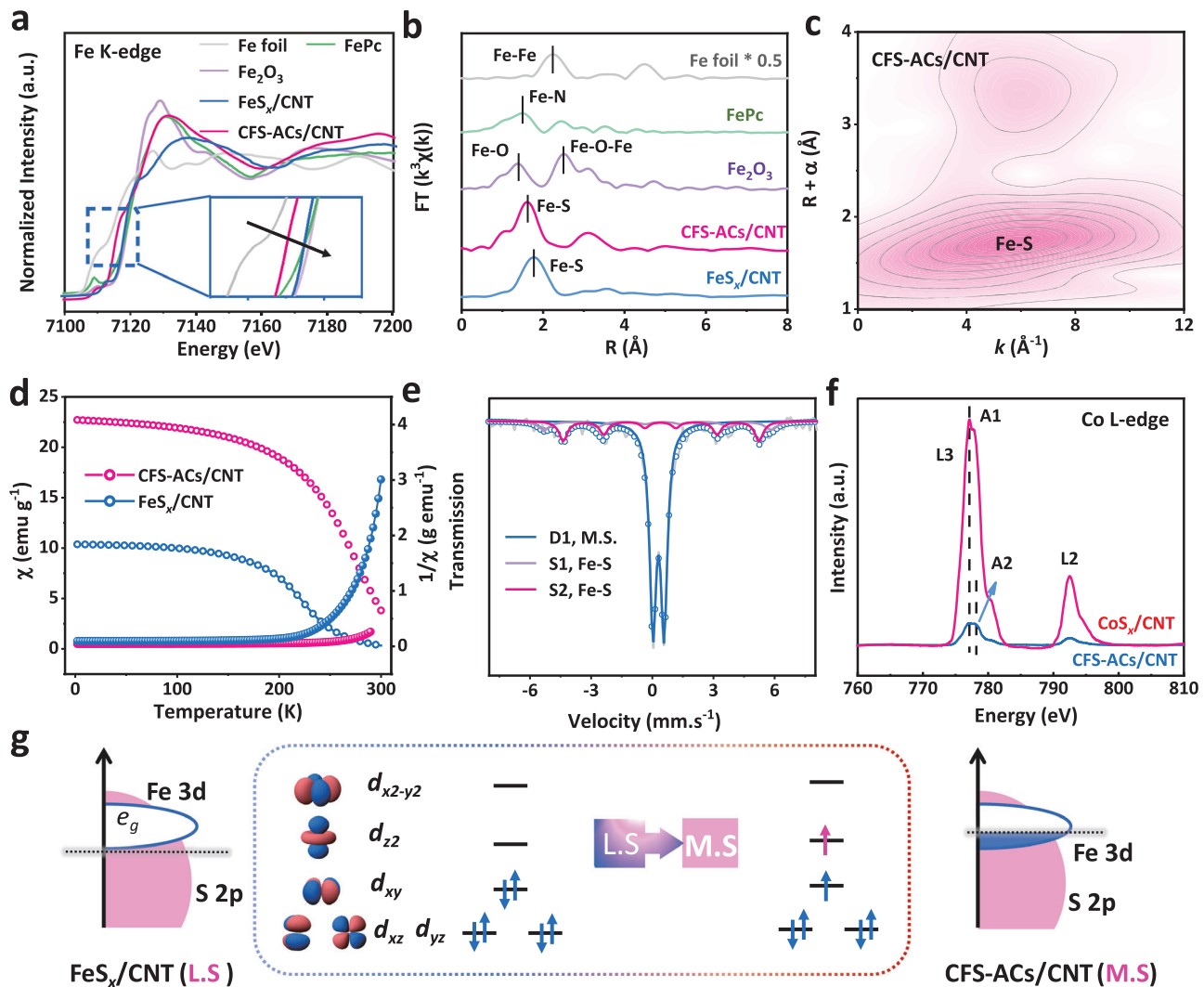

**Fig. 2 | XAS analysis, $^{57}$Fe Mössbauer spectroscopy and magnetic susceptibility.** **a** Fe K-edge XANES and **b** Fourier-transform EXAFS spectra and of CFS-ACs/CNT, FeS$_x$/CNT and CoS$_x$/CNT. **c** Wavelet transforms of the experimental $k^3$-weighted EXAFS spectra of CFS-ACs/CNT. **d** $M-T$ susceptibility χ and reciprocal 1/χ of CFS-ACs/CNT and FeS$_x$/CNT. **e** Room-temperature $^{57}$Fe Mossbauer spectrum of CFS-ACs/CNT. **f** Co L-edge XANES spectra of CFS-ACs/CNT and FeS$_x$/CNT. **g** Schematic illustration of the transfer of spin-state of the high OER activity for CFS-ACs/CNT electrode (Left is d-orbital occupancy of FeS$_x$/CNT, right is CFS-ACs/CNT).

implied more Fe-$3d$ unpaired electrons. Generally, the changes in orbital occupation and the charge redistribution are accompanied by the transformation of the Fe-$3d$ electron spin configuration. Therefore, to identify the possible variation of the electron spin state, magnetic behavior (Fig. S16) was probed. After the formation of sulfides in the hydrothermal process, the dependence on the magnetic fields in CFS-ACs/CNT and FeS$_x$/CNT samples showed a significant hysteresis phenomenon. Compared with FeS$_x$/CNT, CFS-ACs/CNT catalyst with greater saturation magnetization (Ms) had more unpaired electrons and changed the internal electronic configuration of the metal, thus correcting the local spin state[14,31]. The moderate unpaired electrons enabled optimal $d-p$ orbital interactions between M-$3d$ and the O-$2p$ orbitals, facilitating adsorption/desorption of the oxygen intermediates from the ferromagnetic catalyst CFS-ACs catalyst during the OER rate-limiting step, thus accelerating the OER process kinetics[22,32]. Subsequently, the electron spin configuration of CFS-ACs/CNT was determined with a temperature-dependent magnetic susceptibility (M-T) measurement. Figure 2d and Fig. S17 show that paramagnetic-ferromagnetic phase transitions occured at the Curie temperatures ($T_c$) for all compounds. Compared with that of FeS$_x$/CNT, the generation of CFS-ACs/CNT decreased the paramagnetism state of Fe species, and shifted $T_c$ to higher temperatures while generating a significant

magnetic response[31,33], which implied the enhanced disorder and more free electron travel around the Fe species. From the Curie-Weiss law fitting, the effective magnetic moment ($\mu_{eff}$) for CFS-ACs/CNT was calculated as 2.45 μ$_B$. Then, the number of unpaired $3d$ electron ($n$) was determined for the Fe(II) site according to the equation: $\mu_{eff} = \sqrt{n(n+2)}$, whereby $n$ was ~2 for CFS-ACs/CNT and higher than that of FeS$_x$/CNT (~0)[34]. $^{57}$Fe Mössbauer spectroscopy (Fig. 2e and Table S2) was conducted to verify the localized spin states of a single the Fe sites in multi-metallic ferromagnetic catalysts. The main peak was fitted with one doublet of D1 (blue), assigned to M.S Fe(II)[35]. Therefore, there were no excess unpaired electrons assigned to the Co site. It is reasonable to assume the existence of Co in a low-spin state ($t_{2g}^6 e_g^0$), which has unfilled $e_g$ orbitals and strong bonding with the adsorbed OH* intermediates. Furthermore, Co L-edge near edge X-ray absorption fine structure (NEXAFS) spectroscopy was used to investigate the intensity of Co $3d$ empty/partially filled electronic state of CFS-ACs/CNT and CoS$_x$/CNT. As shown in Fig. 2f, two peaks (A1 and A2) for the Co L$_3$ edges were dominated by enhanced "white line" features that implied unoccupied $t_{2g}$ and $e_g$ orbitals, respectively. The peak A1 was assigned to a transition from the Co 2$p_{3/2}$ to the $3d_{z2}$ orbital, while peaks A2 indicated a transition to the $3d_{x2-y2}$ orbital[36,37]. The unoccupied Co $e_g$ orbitals of the CFS-ACs/CNT and CoS$_x$/CNT

samples were directly reflected by the absorption intensity ratio of the $L_3/L_2$ edge, which largely reflects the spin ground state generated by the crystal field effect[38]. $CoS_x/CNT$ displayed a larger $L_3/L_2$ intensity ratio of 3.5, while that of CFS-ACs/CNT was 2.5. Therefore, suppression of the splitting energy of $CoS_x/CNT$ suggested a weak crystal field effect that favored higher $e_g$ orbital occupancy of the Co center. Combined with $M-T$ curves, $CoS_x/CNT$ catalyst exposed a unfilled $e_g$ orbitals occupancy, called low spin state ($Co^{3+}$ $t_{2g}^6e_g^0$). Not surprisingly, This was consistent with our initial reasonable assumption that Co atoms of CFS-ACs/CNT had unfilled $e_g$-orbitals. Yang Shao-Horn et al[4]. proposed that the optimal $e_g$-orbitals occupancy for binding the OER intermediates was almost 1.2. Thus, compared with single Fe site with $3d^6$ ($t_{2g}^6e_g^0$) configuration, CFS-ACs/CNT exposed optimal $e_g$-orbitals occupancy (Fig. 2g). We expected that the synergism effect of local configuration of $Fe^{2+}$ M.S state ($t_{2g}^5e_g^1$) and $Co^{3+}$ L.S state ($t_{2g}^6e_g^0$) would lead to an electronic structure with optimal adsorption energy for OER intermediates, resulting in higher OER catalytic activity.

## OER electrocatalytic performance of CFS-ACs

The OER performance of CFS-ACs/CNT was examined in $O_2$-saturated 1.0 M KOH solutions with a typical three-electrode system. For comparison, CFO-p/CNT, $FeS_x/CNT$, $CoS_x/CNT$, bulk $CoFeS_x$, clean CNTs and commercial $IrO_2$ were also measured. All prepared catalysts were configured into a homogeneous ink and added to the surface of Ni foam (NF) substrate in a uniform drop (600 μL) to form working electrode (Fig. S18). The standard samples (CFS-ACs/CNT) were prepared by a series of content controls to obtain the optimal ratio of Co, Fe and S (Fig. S19). The steady-state linear sweep voltammetry (LSV) shown in Fig. 3a. CFS-ACs/CNT exhibited excellent OER activity with a lower overpotential of 270 mV at 20 mA cm⁻², which was 100 mV and 80 mV less than those of commercial $IrO_2$ and $FeS_x/CNT$, and presented the smallest Tafel slope (Fig. S20 and Fig. 3b) value of 77.6 mV·dec⁻¹. Additionally, LSV curves with $iR$ compensation of CFS-ACs/CNT, $FeS_x/CNT$, $CoS_x/CNT$ and pure CNT are provided in Fig. S21a–e. Similarly, CFS-ACs/CNT also showed superior OER activity than others prepared samples. To rule out the

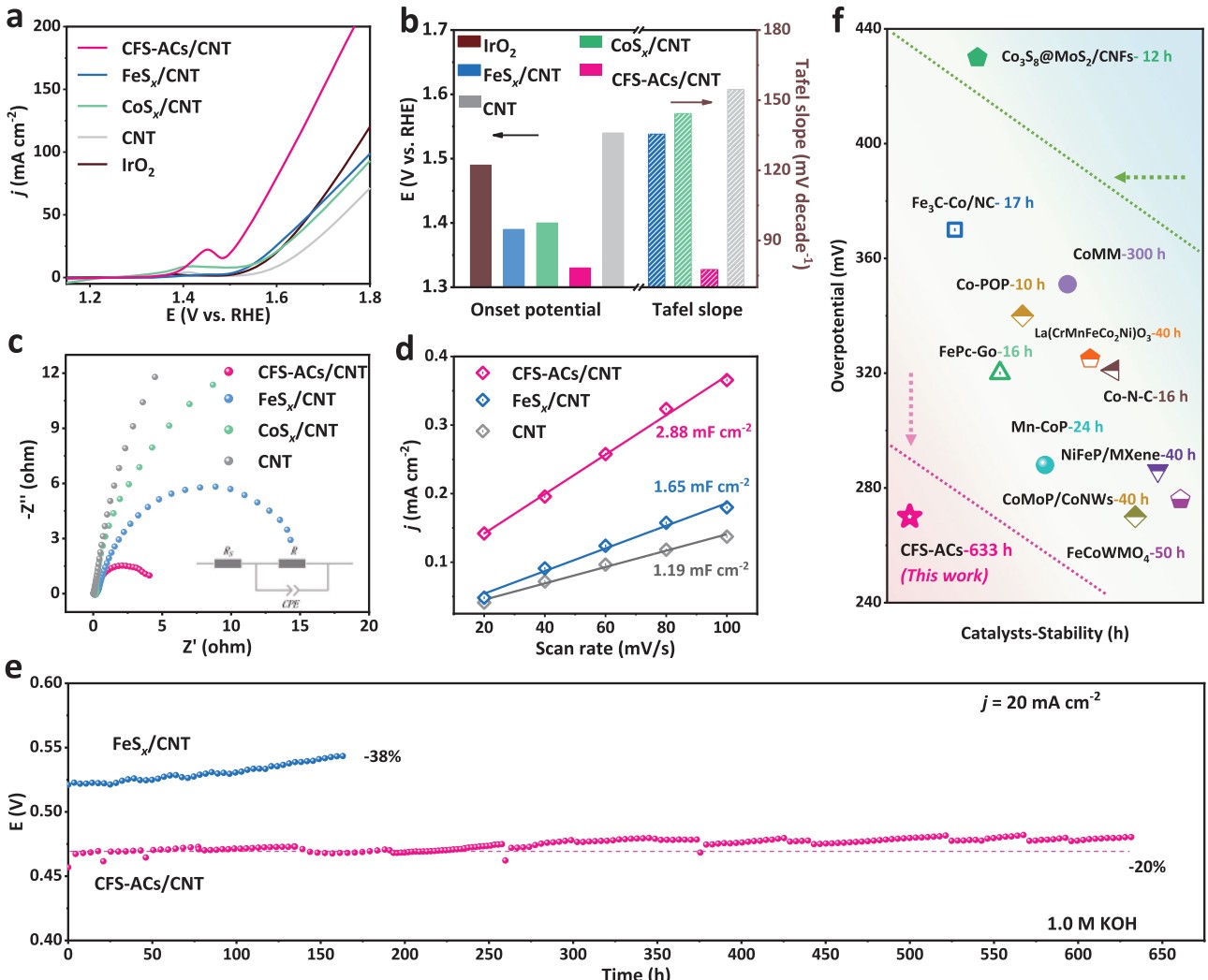

**Fig. 3 | OER performance measurements. a** LSV curves of CFS-ACs/CNT, $FeS_x$/CNT, $CoS_x$/CNT, pure CNT, commercial $IrO_2$ and Ni foam substrate without $iR$ compensation, measured by coating the catalyst on Ni foam (1*2 cm²) with the mass loading of 3 mg·cm⁻². **b** Activity comparison of onset potential at 20 mA cm⁻² and Tafel slope for of CFS-ACs/CNT (red), $FeS_x$/CNT (blue), $CoS_x$/CNT (green), $IrO_2$ (brown) and pure CNT (gray). **c** EIS spectra of CFS-ACs/CNT with a smaller quasi-semicircle ($R_{ct}$ about 4.12 Ω), $FeS_x$/CNT with a quasi-semicircle ($R_{ct}$ about 14.32 Ω), $CoS_x$/CNT and pure CNT. **d** double capacitive current ($C_{dl}$) of CFS-ACs/CNT, $FeS_x$/CNT and pure CNT. **e** The chronopotentiometry curve of $FeS_x$/CNT and CFS-ACs/CNT in 1.0 M KOH (pH = 14). **f** Comparison of the activity and stability of different transition metal catalysts using different symbols to represent them. The red areas represent catalysts with a good balance of activity and stability, the yellow areas are moderate, and the green areas are weak. The references to the data points are supplied in Supplementary Table 4.

influence of NF substrate, the pure NF sample that underwent a similar procedure was also evaluated, and exposed a negligible OER activity (Fig. S22a). Meanwhile, the OER acitivity of CFO-p/CNT was also evaluated (Fig. S22b). Similarly, the OER performance of bulk CoFeS$_x$ was estimated via LSV curve and current density plot with CFS-ACs/CNT and FeS$_x$/CNT at 1.45–1.85 V (Fig. S23) to highlight the effectiveness and foresight for the rational design of cluster structure. To further assess the charge transfer, the Nyquist plots measured by electrochemical impedance spectroscopy (EIS) of CFS-ACs/CNT, CoS$_x$/CNT, FeS$_x$/CNT and CNT were shown in Fig. 3c. The result sample exhibited a smaller charge transfer resistance (R$_{ct}$), a faster kinetics, which coincided with the lower Tafel slope value[39]. The electrochemical surface areas (ECSAs) of as-prepared catalysts were evaluated through the electrochemical double-layer capacitances (C$_{dl}$) in 1 M KOH which were acquired by cyclic voltammetry (CV) at different scan rates (the detailed experiments in Supplementary Fig. 24 and Fig. 3d). The optimal C$_{dl}$ of CFS-ACs/CNT indicated more electrochemically accessible surface active sites, which coincided with the optimal cluster structure. Meanwhile, the ECSA-corrected LSV curve (Fig. S25) obtained after normalizing the current density using ECSA showed that CFS-ACs/CNT had the highest intrinsic activity. Insufficient stability is another major problem for most highly active and highly dispersed OER catalysts. First, corrosion resistance was characterized by CV cycling in Fig. S26. There was no distinct decay before and after 2000 CV cycles over the potential range 1.0 - 1.7 V vs. RHE about the overpotentials, which implied the high corrosion resistance for CFS-ACs/CNT. As shown in Fig. 3e and Fig. S27, chronopotentiometry measurement of CFS-ACs/CNT and FeS$_x$/CNT in 1 M KOH were carried out. CFS-ACs/CNT exposed strong electrochemical stability with continuous OER test run for 633 h without a distinct potential drop, while FeS$_x$/CNT has shown a significantly increasing overpotential after ~180 h continuous chronopotentiometry measure with a potential drop of 38%. Meanwhile, i–t curves after iR compensation of CFS-ACs/CNT, FeS$_x$/CNT and CoS$_x$/CNT at high current density (100 mA cm$^{-2}$) also provided to further estimate their electrochemical stability. As shown in Fig. S21f, after the 70 h continuous measure at high current density, the current density of CFS-ACs/CNT catalyst only decreased by 4 mV. Furthermore, to validate the structural stability provided by sulfur atoms, chronopotentiometry measurements was also carried out for CFO-p/CNT in 1 M KOH. As shown in Fig. S28, the CFO-p/CNT catalyst exhibited weak stability, with a 40% drop in current density after constant 70 h of constant OER testing, which further demonstrated that the introduction of S optimized the electronic structure of the catalyst and successfully attenuated the aggregation state of CoFeS$_x$ clusters during the catalytic process. Consistent with previous reports[37], oxidation state of Co and Fe species were susceptible to changes during the OER. Therefore, XPS and Raman spectra were accomplished to analyze the structure or species evolved after the OER process. After the potential was applied for a period of time, the surface on the catalysts was reconstituted to produce amorphous Fe oxyhydroxides (Fig. S29, 30 and detailed Supplementary Notes), and Fe species were oxidized to +3 valence state. Meanwhile, after stabilization, the O atom content increased with the decreases of active species Fe/Co, while intensity of the S 2p peak for CFS-ACs/CNT only showed slight degradation of S content from 19.01–14.8 at % (Table S3), which further demonstrated the structural stability after the introduction Co, S atom. Furthermore, CFS-ACs/CNT outperformed those of the best catalysts previously reported in literature (Table S4), in terms of both electrochemical activity and stability (Fig. 3f). The comparison of OER activity and stability of CFS-ACs/CNT with those of related single atom CoFe−N$_x$, oxides and other transition metal-based catalysts denoted the effect of ACs structure and coupling mechanism.

## Exploration of the catalytic mechanism from spin state perspective

To further investigate the relationship between spin state of active species transformation from the viewpoint of electronic transition during OER process, in-situ XAS and Raman measurements is also performed. The XANES spectra shown in Fig. 4a, revealed that a conspicuous absorption peak located at 7715 eV gradually shifted toward higher energy with the promotion of the applied potential, which signified an oxidation state change during the OER process. During the oxidation process, the disappearance of Fe$^{2+}$, Fe sites were gradually oxidized to Fe$^{3+}$, which coincided with XPS spectra after long-term measurement (Fig. S30). In Fig. 4b, with the increase in applied potential from the OCP, 1.245 V–1.445 V, the main peak progressively shifted in the low-energy direction, which suggested a transformation in the Fe-S coordination environment and changed in the electronic structure transition of Fe during the OER process[40]. When the potential reached to 1.445 V, the peak was located at -1.56 V, close to that of Fe-O coordination environment[3]. Fe-O coordination indirectly proved that CFS-ACs gradually transfered to the real active phase metal oxyhydroxides during the OER process, and this was accompanied by gradual disappearance of Fe-S bonds[37,41,42]. In-situ Raman spectroscopy further verified the evolution of the active species during electrochemical measurement (Fig. 4c and Fig. S31). The Raman spectrum at the initial potential of 1.23 V showed only two main peaks at 344 and 386 cm$^{-1}$, which were attributed to $E_g$ and $A_g$ modes of Fe-S stretching in CFS-ACs/CNT, respectively[43]. As the potential increased, the intensity of the $A_g$ peak decreased significantly, indicating that a phase transition occured. A band at appeared at 463 cm$^{-1}$ with a potential of 1.43 V and disappeared above 1.73 V, while a band at ~480 cm$^{-1}$ emerged at 1.73 V (Fig. S31). Both bands were assigned to iron oxide and oxyhydroxide, although many accompanying bands were absent. Thus, these two bands were likely associated with the Fe-O vibration modes of poorly crystalline phases formed during the surface oxidation. Moreover, CoO$_2$ active phase belong to Co$^{4+}$-O* was not found[44].

In general, charge redistribution and transfer are accompanied by a leap in the 3d electron spin configuration. As shown in Fig. 4d, CFS-ACs/CNT displayed a larger L$_3$/L$_2$ intensity ratio of 1.6, while that of CFS-ACs/CNT after stability was 1.3. Therefore, the suppression of the splitting XANES structure of CFS-ACs/CNT suggested a weak crystal field enableing a higher $e_g$ orbtial occupancy of the Fe center. The Fe$^{2+}$ site of the CFS-ACs/CNT catalyst exhibited a half-filled d$_{z2}$ orbital occupancy with a medium spin state ($t_{2g}^5 e_g^1$). Therefore, because CFS-ACs/CNT before and after OER stability showed approximate intensity ratio, Fe$^{3+}$ should have a similar $e_g$ orbital occupancy of ~1. Similarly, as shown in Co L-edge XANES (Fig. S32), CFS-ACs/CNT displayed a smaller L$_3$/L$_2$ intensity ratio than CFS-ACs/CNT after stability. Co$^{3+}$ site of CFS-ACs/CNT catalyst had unfilled $e_g$ orbitals with the low spin state ($t_{2g}^6 e_g^0$). Therefore, Co$^{3+}$ site after stability may have partially filled $e_g$ orbitals with the medium spin state ($t_{2g}^5 e_g^1$) due to smaller splitting energy[45]. According to the bond order theorem and orbital interactions between cations and the oxygen intermediates in Fig. 4e, Fe$^{2+}$ M.S. state ($t_{2g}^5 e_g^1$) exhibited poorer spin-orbit coupling interactions than the Co$^{3+}$ L.S. state ($t_{2g}^6 e_g^0$), implying a weaker OH* adsorption capacity. In recent years, numerous investigations have been devoted to revealing the relationship between catalytic activity and the spin configurations of catalysts, among these, the ferromagnetic element Fe has been the most studied (Table S5). Hyung-Suk Oh et al[37]. proposed the evolution of the spin state of Co$^{3+}$OOH site from L.S. ($t_{2g}^6 e_g^0$) to M.S. ($t_{2g}^5 e_g^1$) at the applied potential of 1.4 V. Similarly, after oxidation process, Fe$^{2+}$ was oxidized to Fe$^{3+}$ due to the generation of Fe oxyhydroxides but the M.S. state maintains, at which point the spin state of Co$^{3+}$ changed from L.S. to M.S., but the valence state remained unchanged. The Fe$^{3+}$ M.S. showed 1 of $e_g$-orbital occupancy, possessed slightly strong O* adsorption energy for the OER. Supporting this in

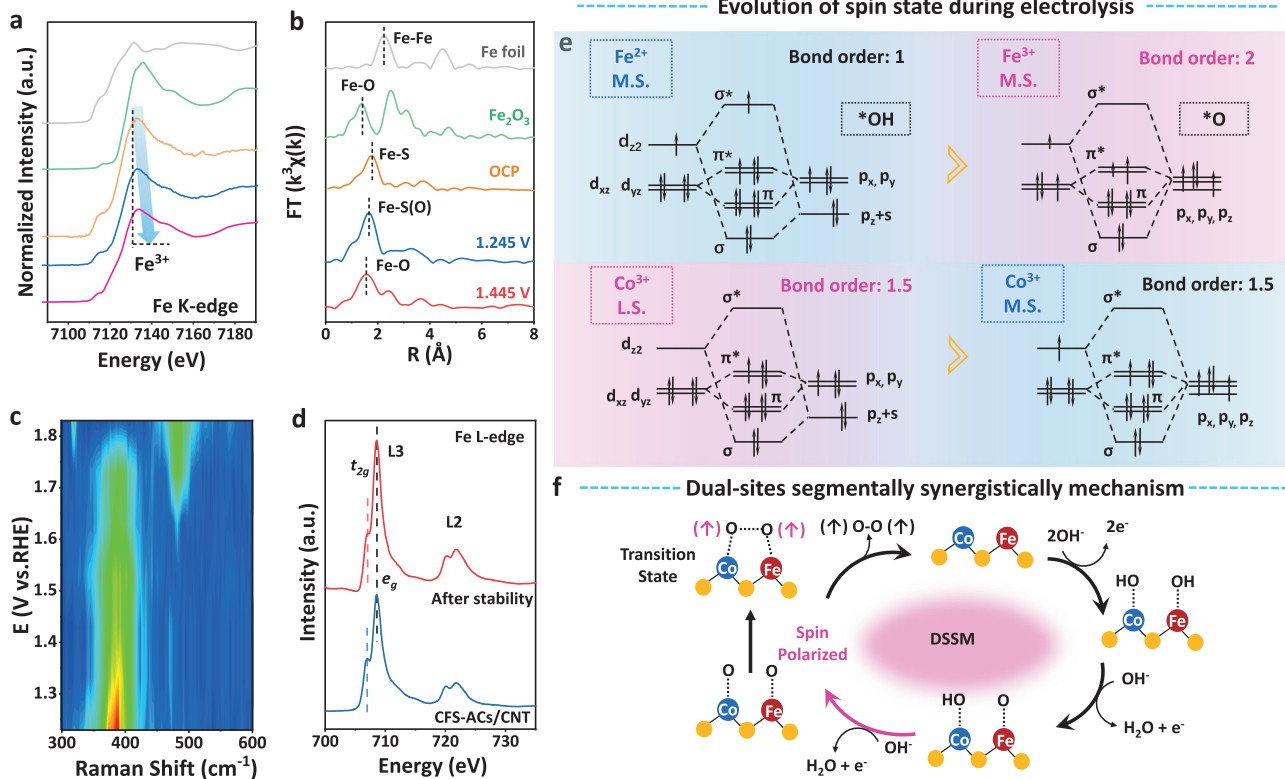

**Fig. 4 | Spin state analysis during OER process. a** Fe K-edge XANES spectra, **b** Fe K-edge FT-EXAFS during OER process with the different potentials (*vs* RHE) of OCV, 1.245 V, and 1.445 V in 1 M KOH, respectively. **c** In-situ Raman spectroscopy of CFS catalyst in 1.0 M KOH. **d** Fe L-edge XANES spectra before and after stability of CFS-ACs/CNT. **e** The orbital interactions between cations and the OER intermediates of different systems, Zhang et al. regulated the L.S. to M.S. ($t_{2g}^4 e_g^1$) spin state transformation of $Fe^{3+}$ with the introduction of $Mn^{[46]}$ to modulate the occupy of antibonding π-orbital of oxygen. Similarly, Wang et al. enhanced the activity of oxygen reduction reaction (ORR) through optimizing $O_2$ adsorption of $Fe^{3+}$ M.S ($t_{2g}^4 e_g^1$) site by axial Fe-O-Ti ligand[47]. Herien, $Fe^{3+}$ sites (M.S., $t_{2g}^4 e_g^1$) showed the optimal O* adsorption, when $Co^{3+}$ sites (L.S., $t_{2g}^6 e_g^0$) demonstrated dominant *OH adsorption during OER process. Therefore, electrophilic OH* intermediates of Lewis basic $OH^-$ were preferentially bound at the Fe sites and lost thereby losing electrons to produce Fe-O* intermediates. From the valence or spin state changes at different sites during oxidation, we expected that it enables effective functional differentiation of active species in the proton-coupled electron transfer steps and simultaneous segmentation to promote OER kinetics. Combining a schematic representation of the spin-orbit interaction of the metal with the oxygen intermediate and the valence change of the metal site at the applied potential, a feasible CoFe dual-site OPM mechanism (i.e., the DSSM mentioned above) is proposed: (Fig. 4f): a dual-site segmentally synergistic catalysis mechanism enabled the strong OH* adsorption capacity at the Co site and the strong O* adsorption capacity of the Fe site, this permitted the simultaneous presence of Co-O* species with Fe-O* and triggered the switch of the O-O coupling mechanism, therefore stimulating the formation of the adsorbed Co-O-O-Fe intermediate.

A single-domain ferromagnetic catalyst $CoFeS_x$ stimulated the presence of an electric field inside the catalyst and induced spin-polarized water oxidation to key oxygen intermediates, which was contributing to the parallel spin to reduce the formation barrier of O − O coupled intermediates and thus facilitated the direct desorption of triplet oxygen (↑O = O↑)[19]. Therefore, scaling relationship between OH* and OOH*, which constrains the OER adsorption evolution, was

Fe and Co during OER process according to the bond order theorem. The red region represents more favorable metal spin-orbit coupling interactions. **f** The schematic diagram of dual-site adsorbate evolution mechanisms for coupled O-O bonding during the OER (blue for Co atoms, red for Fe atoms, yellow for S atom).

broken while avoiding the poor structural stability caused by the traditional O-O coupling mechanism[48].

## DFT calculation

Furthermore, to evaluate whether Co-O-O-Fe coupling mechanism would afford the best OER performance, DFT calculations were performed. As a comparison, typical AEM mechanisms and iron sulfide (single site) models were also considered. The $OH^-$ rich electrolyte was accompanied by a long substitution process of the active phase, so we choose the transiently stable intermediate phase CoFeS(O) produced by partial substitution for the analysis of the OER mechanism. Therefore, OER process was simulated with spin-polarized DFT calculations by applying DFT + U method to study the OER activities of the CoFe dual-site with different mechanisms. Figure S33−37 shows 4e⁻ pathway for typical AEM mechanism and optimal OPM mechanism with CoFe as dual-site active species. Furthermore, the spin polarization spatial distribution of Co (Fe) site with oxygen intermediates was calculated to evaluate the spin exchange interactions of the itinerant electrons near Co-S-Fe(O) region (Fig. S38 and Fig. 5a). The spin-selective electron transfer from the single-linear state $OH^-$ or $H_2O$ to triplet-state oxygen was necessary to guarantee the smooth operation of the OER 4e⁻ transfer process. However, this is a spin-forbidden process. For ferromagnetic electrocatalysts, electron-electron repulsion are small when the short-range quantum spin-electron couples with the adsorbed oxygen (reactant), thus inducing spin-dependent conductivity and lowering the rate-limiting bond energy[14,20,49].

Meanwhile, to assess the impact of surface spin states on adsorbate adsorption energy in DFT calculations, particularly concerning magnetic materials, two Co-Fe surfaces were introduced for comparison: one with a distinct spin moment (ground state spin moment) and the other non-spin polarized, achieved by setting the spin to zero in the

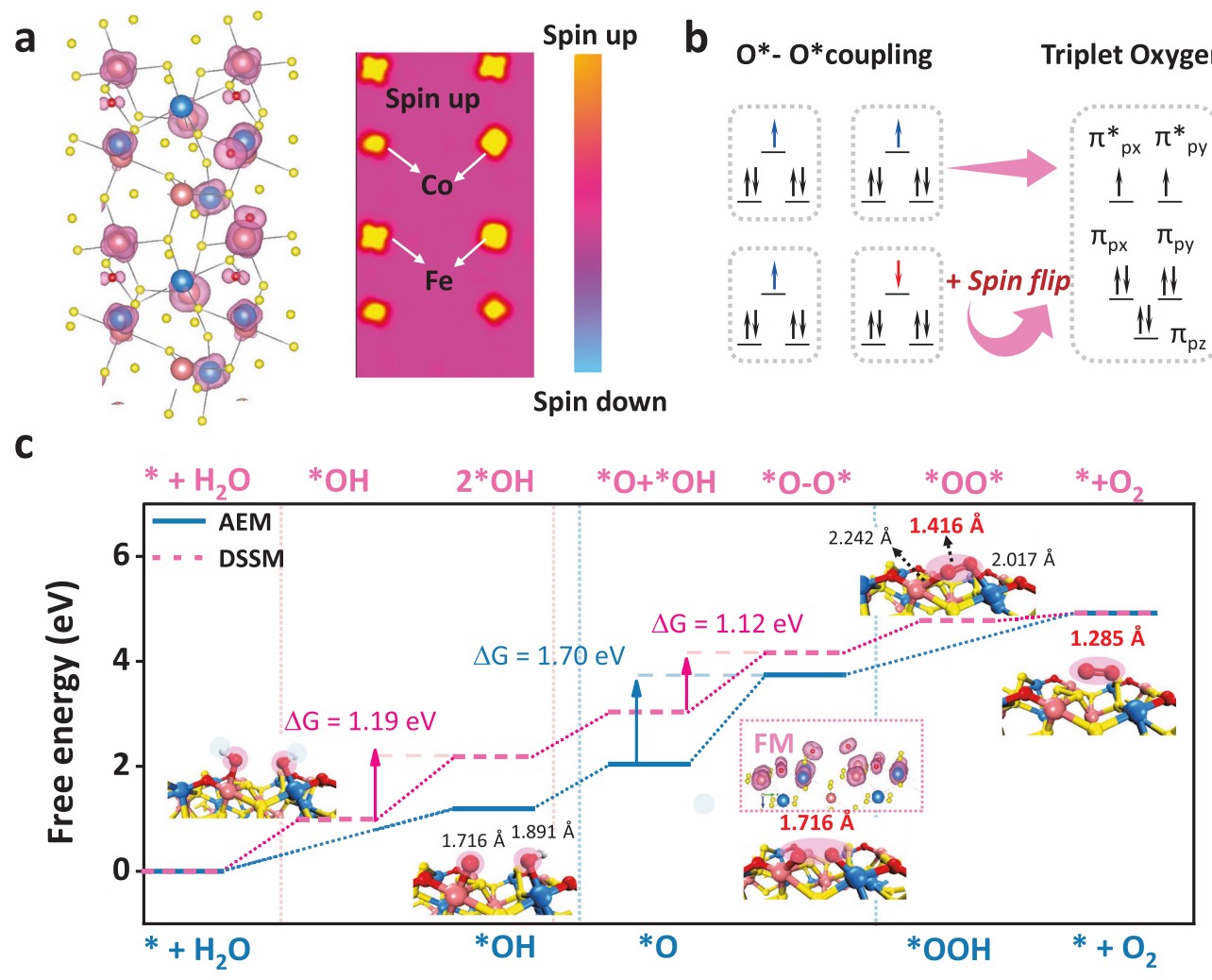

**Fig. 5 | OER mechanism analysis based on DFT calculations. a** Spin density and planar distribution maps of the spin polarization of the key oxygen intermediates (Co(O*)-Fe(O*)). The pinkish purple isosurfaces represent the spin-up states. **b** The parallel arrangement of spin electrons in the oxygen-oxygen coupling of adjacent metal sites facilitates triplet oxygen production. **c** The free energy diagram (ΔG) of typical AEM mechanism and DSSM pathway of OER including all oxygen intermediates OH*, O*, OOH* and *OO*. Insets show the spin density plot of Co(*O)−Fe(O*) over the ferromagnetic CFS model towards the transition from *O to O₂. Blue balls for Co atoms, pink balls for Fe atoms, yellow balls for S atoms, red balls for O atoms and white balls for H atoms.

system and recalculating the lattice constant[50]. As shown in Fig. S39, for a range of adsorbates, non-spin polarized surfaces showed stronger the adsorption energy than that of spin-polarized ground state surfaces, especially in RDS step. In Fig. 5a, the key intermediate state Co(O*)−Fe(O*) revealed the ferromagnetic coupling between these CoFe dual-site, which implied that the spin-parallel alignment of the oxygen intermediate can effectively accelerate the desorption step (release of the triplet-state O₂). As illustrated in Fig. 5b, due to anti-ferromagnetic coupling, the application of additional energy was required to overcome the pairwise energy barrier and induce the spin-unaligned alignment, which can cause spin flips and achieve a final spin-parallel alignment to release the triplet state oxygen. Meanwhile, the trend of ferromagnetic coupling of O* further demonstrated the preference of O-O coupling rather than linear constraints on OOH* species and the efficiency of oxygen release, which is a feature of the OPM mechanism[12,13]. Moreover, Fig. 5c shows the Gibbs free energy diagrams of two different pathways: typical AEM and DSSM mechanism. As plotted in Fig. 5c and S37, the rate determining step (RDS) was calculated to be the formation of *OOH or *O-O* intermediates from the step of *O for both CFS-ACs/CNT and FeSₓ/CNT, indicating the

stronger binding of *O on Fe site, which was consistent with the established scaling relationship. The RDS barrier for DSSM pathway was 1.12 eV, showing almost great advantage compared with that of AEM pathway. As excepted, the two O atoms exhibited parallel spin arrangements at the ferromagnetic-coupled active sites, which was more favorable for desorption to form the spin triplet state O₂.

## Discussion

In summary, we fabricated a small-sized (1.5 ~ 3 nm) CFS-ACs catalyst tied to CNT by a low-temperature hydrothermal technology, which exhibited excellent OER activity with the overpotential of 270 mV at 20 mA cm⁻² and remarkable durability (633 h without potential loss) in 1 M KOH. According to magnetic characterization and ⁵⁷Fe Mössbauer spectroscopy, NEXAFS spectrum, large hysteresis loop and coercivity and proprietary ferromagnetic curve belongs to CFS-ACs/CNT. We distinguished for the first time the functionality and spin-state of different active sites with Fe²⁺ (M.S, $t_{2g}^5 e_g^1$) as well as Co³⁺ (L.S, $t_{2g}^6 e_g^0$). Based on in-situ/*operando* X-ray adsorption measurements and Raman spectroscopy, we proposed a dual-site segmentally synergistic catalytic mechanism based on the evolution of the metal-oxygen

intermediate orbital interactions due to changes in the $e_g$ orbital electron occupation of the metal active site during the OER process, in which Co-O-O-Fe coupling mechanism successfully avoided the limitations of the scaling relationship between the adsorption oxygen mechanism OOH* and OH*. Meanwhile, small-sized (1.5 – 3 nm) ferromagnetic CFS-ACs catalyst can be regarded as a single-domain ferromagnetic catalyst, which stimulated the presence of an electric field inside the catalyst, induced spin-polarized water oxidation of key intermediate oxygen species and generated a spin parallel alignment to reduce the formation barrier of O – O coupled intermediates and thus facilitate the direct desorption of triplet oxygen (↑O = O↑). DFT calculations further revealed that O-O coupling mechanism is feasible and outperformed to typical single-site adsorption mechanism. Our work has not only achieved simultaneous promotion of O-O coupling intermediates generation and $O_2$ release, but also offers a steerable strategy and unique insights into the impact of spin state changes on oxygen adsorption intermediates during OER.

## Methods

### Chemicals

Ferric nitrate ($Fe(NO_3)_3 \cdot 9H_2O$), cobaltous nitrate ($Co(NO_3)_2 \cdot 6H_2O$), thiacetamide (TAA) all were purchased from Beijing Enokai Technology Co.,LTD. Potassium sulfate ($K_2SO_4$) and sodium hydroxide (KOH) were purchased from Sinopharm Chem. Reagent Co., Ltd. Multi-walled carbon nanotube (CNT, >95%, 10 – 20 nm in diameter and 10–30 μm in length) was obtained from Beijing Deke Daojin Technology Co., LTD. Sodium borohydride ($NaBH_4$) was acquired by Tianjin Fengchuan Chemical Reagent Technology Co., Ltd. Nickle foil (NF, 99.8%) was supplied by Suzhou Shengerno Technology Co., Ltd, and Nafion (5.0 wt.%) was obtained from Sigma-Aldrich. All chemicals were used as received without any further purification. The $IrO_2$ catalyst was obtained from Johnson Matthey. Deionized water used in all experiments was ultrapure.

### Synthesis of CoFeOx/CNT (CFO-p/CNT)

Firstly, 70 mg CNT were dispersed in 50 mL ethanol with continuous sonicating for 30 min to form solution A. Secondly, 2 mmol $Co(NO_3)_2 \cdot 6H_2O$ and 2 mmol $Fe(NO_3)_3 \cdot 9H_2O$ were dissolved in 20 mL ethanol under 10 min magnetic stirring to form solution B. Solution B was slowly added dropwise to solution A with continuous stirring for 10 h, called solution C. Subsequently, 10 mL of 1.25 mmol of $NaBH_4$ was slowly added dropwise to solution C in an ice-water environment. Finally, the solution was washed by centrifugation with ice-cold ethanol and ultrapure water, followed by freeze-dried to obtain the CFO-p/CNT precursor. Similarly, $FeO_x$/CNT and $CoO_x$/CNT precursor were prepared with the same process without the addition of Co source and Fe source, respectively.

### Synthesis of CFS-ACs/CNT

80 mg as-prepared CFO-p/CNT precursor was dispersed in 40 mL deionized water under stirring vigorously for 30 min. Then, 25 mL 40 mmol thioacetamide was added dropwise while stirring for 30 min. The resulting solution was transferred into a 100 mL Teflon-lined stainless-steel autoclave, followed by heating at 160 °C for 6 h. Finally, the final CFS-ACs/CNT was obtained by washing and freeze-drying. Similarly, $FeS_x$/CNT and $CoS_x$/CNT were fabricated with the same amount of thioacetamide. Meanwhile, $Co_1Fe_1@S$/CNT, $Co_4Fe_4@S$/CNT, $CoFe@S_{10}$/CNT and $CoFe@S_{50}$/CNT were prepared with the differernt amount of 1 mmol $Co(NO_3)_2 \cdot 6H_2O$ and 1 mmol $Fe(NO_3)_3 \cdot 9H_2O$; 4 mmol $Co(NO_3)_2 \cdot 6H_2O$ and 4 mmol $Fe(NO_3)_3 \cdot 9H_2O$; 10 mmol thioacetamide and 50 mmol thioacetamide.

### Characterizations

The component of all-prepared samples was characterized by X-ray diffraction (XRD) patterns, which were collected in by a Y-2000 X-ray Diffractometer using copper Kα radiation (λ = 1.5406 Å) at 40 kV, 40 mA. The structure of samples was proved by X-ray photoelectron spectroscopy (XPS), which were performed with thermo Scientific K-Alpha spectrometer using a focused monochromatic Al Kα (1486.6 eV) X-ray beam with a diameter 400 μm. Raman spectra were collected on a HORIBA Scientific LabRAM HR Evolution with a 532 nm laser source. The morphology of the samples was characterized by transmission electron microscopes (TEM, FEI Tecnai G220) with an accelerating voltage of 200 kV and field-emission scanning electron microscope (FE-SEM, JEORJSM-6700F). HAADF-STEM images were acquired from JEOL JEM-ARM200F at an accelerating voltage of 200 kV. The Fe K-edge and Co K-edge X-ray absorption near edge structure (XANES) and the extended X-ray absorption fine structure (EXAFS) were investigated at the BL14W1 beamline of Shanghai Synchrotron Radiation Facility (SSRF) in the fluorescence mode using a fixed-exit Si (111) double crystal monochromator. The obtained XAFS data was processed in Athena (version 0.9.26) for background, pre-edge line and post-edge line calibrations. Then Fourier transformed fitting was carried out in Artemis (version 0.9.26). The k3 weighting, k-range of 3–12 Å$^{-1}$ and R range of 1–3 Å were used for the fitting of foils; k-range of 3 - -10 Å$^{-1}$ and R range of 1–2.6 Å were used for the fitting of Fe and Co samples. The four parameters, coordination number, bond length, Debye-Waller factor and E0 shift (CN, R, σ2, $\Delta E_0$) were fitted without anyone was fixed, constrained, or correlated. For Wavelet Transform analysis, the χ(k) exported from Athena was imported into the Hama Fortran code. The parameters were listed as follow: R range, 1–4 Å, k range, 0 - -10 Å$^{-1}$ for Fe and Co samples; k weight, 2; and Morlet function with κ = 10, σ = 1 was used as the mother wavelet to provide the overall distribution. $^{57}$Fe Mössbauer spectra of the catalysts were obtained on a Topologic 500 A spectrometer driven with a proportional counter at room temperature. The incident X-ray beam was monitored by an ionization chamber filled with $N_2$, and the X-ray fluorescence detection used a Lytle-type detector filled with Ar. The EXAFS raw data were then background-subtracted, normalized and Fourier transformed by standard procedures with the IFEFFIT package.

### Synchrotron-based XAS measurements

The Fe and Co L-edge X-ray absorption (XAS) spectra were measured at the Beamlines MCD-A and MCD-B (Soochow Beamline for Energy Materials) at National Synchrotron Radiation Laboratory (NSRL, China). During the measurement, samples were placed at room temperature. Both the Fe and Co L-edge were measured by means of total electron yield (TEY), measuring the drain current as a function of the photon energy. Multiple scans were measured and averaged.

### In-Situ X-ray Absorption Fine Structure (XAFS)

Firstly, the pretreatment as follows: the working electrodes were pretreated by voltammetric cycles between 0.2 and 0.8 V at a rate of 50 mV s$^{-1}$ in $O_2$-saturated 1 M KOH aqueous solution. The catalyst was hand-brushed onto carbon paper (1 × 2cm$^2$). OCV, 1.245 V vs. RHE and 1.445 V vs. RHE were chosen as three representative conditions, the in situ X-ray absorption signals were collected when the working electrode reached a steady state. The electrode was maintained under each condition for 30 min before measuring the spectrum. The Fe K-edge XANES and EXAFS spectra were collected at the Taiwan Photon Source 44 A beamline and BW14 of SSRF. Fe foils was used to calibrate the beamline energy and compare samples. Fluorescence mode detection was performed using a Si (111) double crystal monochromator. An ion chamber detector for samples and the total electron yield was used to measure samples with high concentrations, such as references.

### In-situ Raman

A custom-made two-compartment three-electrode electrochemical cell with a quartz window was used for in situ Raman measurements[51].

2 μL of CFS suspension was dropcast onto the the glassy carbon electrode as a working electrode. A graphite rod was used as the counter electrode. A Ag/AgCl electrode (3.0 M, BASi) was used as the reference electrode. A Gamry Reference 600+ Potentiostat was used to control the electrode potential. Prior to Raman measurement, the glassy carbon electrode was polished by 0.05 um alumina suspension. The Raman spectroscopy measurements were performed using a LabRAM HR Evolution microscope (Horiba Jobin Yvon) equipped with a 532 nm laser, a ×50 objective (NA = 0.55), and a CCD detector. Raman frequency was calibrated using a Si wafer before each experiment. The filter was set to be 50% to keep a low laser intensity to avoid any irradiation-induced modifications of the organic additive deposited surface. The acquisition time was set to 40 s for each spectrum.

## DFT calculation details

All DFT calculations were performed using the Vienna Ab initio Simulation Package (VASP). The spin-polarized DFT calculations were performed with generalized gradient approximation and Perdew–Burke–Ernzerhof (PBE) functional[52] as implemented in the Vienna ab initio Simulation Package[53,54]. The projector augmented-wave method[5] was used to treat the ion-electron interactions. The cutoff energy of 450 eV was set and the self-consistent field (SCF) tolerance was $1 \times 10^{-5}$ eV. Dipole moment corrected was made accordingly in the z direction, and the empirical correction scheme of Grimme[55] was also performed, where the effect of vdW interactions was included explicitly. To describe the strongly-localized interaction from Co-d and Fe−d electrons, the PBEsol exchange-correlation functional was used together with an effective Hubbard parameter within the Dudarev approach with 3 eV for Fe and 3.3 eV for Co, taken from Ref. [56] CoFeS$_4$ (1 1 0) surface was selected with repeated in 4 × 4 unit cell and a vacuum width of 15 Å was used. The bottom two layers were fixed, while the top one layers and the adsorbates was allowed to relax. The 3 * 3 * 1 Monkhorst–Pack mesh k-points was sampled for all surface calculation. And all energy difference of all planes does not exceed 0.02 eV. The VASPKIT package was used for post-processing of the calculation results[57].

The Gibbs free energy calculated by DFT is based on computational hydrogen electrode (CHE) model provided by Nørskov[58], in which the total energy of $H^+/e^-$ is equal to $\frac{1}{2} H_2$ at standard condition ($G_{H^+} + e^- = \frac{1}{2} G_{H_2}$). The free energy of O$_2$ (g) is calculated by

$$G(O_2) = 2G(H_2O) - 2G(H_2) + 4.92 \text{eV} \tag{1}$$

In terms of each elementary step, the Gibbs free energy change is calculated via Eq. (2):

$$\Delta G = \Delta E + \Delta ZPE - T\Delta S \tag{2}$$

where $E$ is the total energy obtained in DFT, $ZPE$ is the zero-point energy, $T$ is the temperature (298.15 K), and $S$ is the entropy. In detail, the $ZPE$ of the adsorbate is calculated by Eq. (3):

$$ZPE = \frac{1}{2} \sum_i h\nu i \tag{3}$$

We determined the entropy of gas-phase molecules from the CRC Handbook[59], the entropy of the adsorbate species are calculated by Eq. (4):[60]

$$S_m^v = R \left[ \frac{\beta hc\tilde{v}}{e^{\beta hc\tilde{v}} - 1} - ln\left(1 - e^{-\beta hc\tilde{v}}\right) \right] \tag{4}$$

where $h$, $c$ and v ($\tilde{v}$) are Plank constant, the speed of light and vibrational frequency respectively.

the contribution from entropy TS is calculated by Eq. (5) with a Harmonic oscillator approximation as:

$$-TS = k_B T \sum i \, ln\left(1 - \exp\left(-\frac{h\nu_i}{k_B T}\right)\right) - \sum i \frac{h\nu_i}{\exp(-\frac{h\nu_i}{k_B T}) - 1} \tag{5}$$

where h, and v are Plank constant and vibrational frequency respectively.

Accordingly, the Gibbs free energy changes for the water oxidation steps using AEM mechanism was calculated using the following Equations:[13]

$$OH^- + M^* \rightarrow M-OH + e^- \tag{6}$$

$$M - OH + OH^- \rightarrow M - O^* + e^- + H_2O \tag{7}$$

$$OH^- + M - O^* \rightarrow M - OOH + e^- \tag{8}$$

$$M - OOH + OH^- \rightarrow M^* + O_2 + e^- + H_2O \tag{9}$$

The intermediates adsorption energy $E_{ads}$ for M-OH, M-O, M-OOH and M-OO can be used DFT ground state energy calculated as:

$$\Delta G1 = G_{M-OH} - G_{M^*} - G_{OH^*} - eU \tag{10}$$

$$\Delta G_2 = G_{M-O} - G_{M-OH} + 0.5 \, G_{H2(g)} - eU \tag{11}$$

$$\Delta G_3 = G_{M-OOH} - G_{M-O} - G_{H2O(l)} + 0.5 G_{H2(g)} - eU \tag{12}$$

$$\Delta G_4 = G_{O2(g)} + G_{M^*} - G_M - OOH + 0.5 \, G_{H2(g)} - eU \tag{13}$$

Moreover, the Gibbs free energy changes for the water oxidation steps using OPM mechanism was calculated using the following Equations:

$$OH^- + M_1^* \rightarrow M_1 - OH + e^-; OH^- + M_2^* \rightarrow M_2 - OH + e^- \tag{14}$$

$$\begin{aligned} M_1 - OH + OH^- \rightarrow M_1 - O^* + e^- + H_2O; M_2 \\ - OH + OH^- \rightarrow M_2 - O^* + e^- + H_2O \end{aligned} \tag{15}$$

$$M_1 - O^* + M_2 - O^* \rightarrow M_1 - OO - M_2 \tag{16}$$

$$M_1 - OO - M_2 \rightarrow M^* + O_2 \tag{17}$$

The intermediates adsorption energy $E_{ads}$ for M-OH, M-O and M-OO-M can be used DFT ground state energy calculated as:

$$\Delta G_1 = G_{M-OH} - G_{M^*} - G_{H_2O(l)} + 0.5 \, G_{H2(g)-eU} \tag{18}$$

$$\Delta G_2 = G_{M-O} - G_{M-OH} + 0.5 \, G_{H2(g)-eU} \tag{19}$$

$$\Delta G_3 = G_{M-OO-M} - 2G_{M-O} \tag{20}$$

$$\Delta G_4 = G_{O2(g)} + G_{M^*} - G_{M-OO-M} \tag{21}$$

## Electrochemical measurements

Typically, all the electrochemical tests were performed in a three-electrode system at an electrochemical station (CHI Electrochemical Station (Model 760 E)) at a room temperature. Ag/AgCl (in 3.0 M KCl) was used as the reference electrode, and a carbon rod as the counter electrode. Firstly, catalyst ink was prepared by dispersing 5 mg of catalyst into 1 mL of water/isopropanol (v/v = 3:7) solvent containing 40 μL 5 wt% Nafion and sonicated for 30 min to form a homogeneous solution. Then, 600 uL homogeneous solution was loaded on a clean NF ($1 \times 1 cm^2$), vacuum dried and used as working electrode. The final loading for all catalysts on the NF was about $3 mg/cm^2$. During oxygen evolution reaction (OER) measurement process, linear sweep voltammetry (LSV) was tested in 1.0 M KOH with a scan rate of $5 mV s^{-1}$. The electrolyte is freshly prepared for each electrochemical measurement. Meanwhile, all the electrochemical test is performed in an H-type electrolytic Teflon cell. All potentials are referenced to the reversible hydrogen electrode (RHE) by the Nernst equation:

$$E(RHE) = E(Ag/AgCl) + 0.197 V + 0.0591 pH \qquad (22)$$

and the overpotential of OER was calculated according to the equation:

$$\eta = E(RHE) + 1.23V \qquad (23)$$

The electrochemical impedance spectroscopy (EIS) was obtained at the open-circuit voltage from 10000–0.1 Hz. The long-term stability of the samples was carried out by the chronopotentiometry method with a stable current density of $20 mA cm^{-2}$. To determine the electrochemical surface area (ECSA) can be obtained by the following equation: $ECSA = C_{dl}/C_s$. $C_s$ is the specific capacitance in 1 M KOH ($C_s = 0.040 mF$) and $C_{dl}$ is the electrochemical double layer capacitance measured using cyclic voltammetry curves (CV, tested in 0.15–0.25 V ranging from $10–50 mV s^{-1}$). CV activation was performed prior to each electrochemical test.

## Data availability

The data that support the plots are available within this paper and its Supplementary Information. All other relevant data that support the findings of this study are available from the corresponding authors on reasonable request. Figures 2–5, Supplementary Figs. 1, 5–7, 8–13, 15–32 and 37 data generated in this study are provided in the Source data files. Source data are provided with this paper.

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

## Acknowledgements

This work was financially supported by the key projects of the National Natural Science Foundation of China (U22A20107, J.N.Z.), and the key projects of the Henan Provincial Science and Technology R&D Program Joint Fund (222301420001, J.N.Z.), the Distinguished Young Scholars Innovation Team of Zhengzhou University (32320275, J.N.Z.), Higher Education Teaching Reform Research and Practice Project of Henan Province (2021SJGLX093Y, J.N.Z.). Acknowledge the Beijing Synchrotron Radiation Facility (BSRF) 1W1B station for XAS measurements. Li-Rong Zheng is grateful to the support of XAS. Acknowledge the B.-J.Xu's group of College of Chemistry and Molecular Engineering, Peking University Beijing for In-situ Raman spectroscopic testing. Acknowledge the Beamlines MCD-A and MCD-B (Soochow Beamline for Energy Materials) at National Synchrotron Radiation Laboratory (NSRL, China).

## Author contributions

J.-N.Z. supervised this research. and S.-R.X. conceived the project and designed the work and S.-R.X., M.-L.L. performed the experiments, including materials synthesis, characterizations, and electrochemical measurements. And K.-Y.Z., B.-J.X. performed In-situ Raman characterizations. S.-H.F., C.W. provided Synchrotron-based XAS measurement. Y.Y. provided theoretical computational support and D.-P.X., J.-N.Z. reviewed and edited the manuscript. All authors have approved the final version of the manuscript.

## Competing interests

The authors declare no competing interests.
