## [Peer review file · Nature Communications]

REVIEWER COMMENTS

Reviewer #1 (Remarks to the Author):

The authors report an unconventional Co-Fe dual-site synergistic mechanism (DSSM) on a single-domain ferromagnetic catalyst CoFeS_x nanoclusters on carbon nanotube (CNT) catalyst (CFS-ACs/CNT), which can effectively break the scaling relationship without sacrificing stability. The authors claimed two different types of sites synergistically promote the production of Co-O-O-Fe intermediates, thereby benefiting accelerating the release of O₂. And various experiments and characterization have been conducted to support above conclusion. Therefore, I would like to recommend this manuscript to be accepted after the following considerations are fully addressed.

1. In Figure 1d, what is the specific crystal facet of lattice fringe in STEM image.
2. Author conduct VSM and ⁵⁷Fe Mossbauer spectrum to confirm the spin state of Fe atoms, how about Co atoms and more experiment should be conducted to analyze the spin configuration of Co atoms.
3. The current density normalized with the ECSA should also be provided to reveal what is major contribution on current density increase.
4. One major concern is the chemical stability in alkaline solution during OER process, whether the catalyst is transferred to other phase after oxygen electrocatalysis? In the current work, the author fails to discuss this matter as no structural characterization was conducted nor presented after the durability test after OER.
5. The authors need to provide the fitting model for the calculation of the coordination number in EXAFS fitting.
6. The equivalent electrical circuit to fit the EIS should be also provided.

Reviewer #2 (Remarks to the Author):

The paper describes studies of Co-Fe dual metal-based catalysts supported on sulfur modified carbon nanotubes for OER in alkaline solution. This field of research has received much attention in the past and represents an important system for understanding the mechanism of the OER reaction. The authors describe samples prepared by thermo-chemical treatment of carbon nanotubes while also proposing an OER pathway based on oxide path mechanism (OPM). Quite interesting results were obtained through diverse characterization techniques. However, some of the points proposed by the authors are not very convincing and/or misleading. Many points in this work should be addressed, as presented in detail below, for publication in Nature Communications to be justified.

1. There are diverse kinds of ferromagnetic catalysts other than the combination of cobalt and iron. Is there a specific reason for choosing cobalt and iron as active phase metals in this work?
2. In EXAFS analysis (Figure 2b and 4b, for example), there should be no distinctive peaks in the range of < 1 angstrom in R space. Perhaps the authors should reconsider the EXAFS fitting conditions.
3. Electrochemical impedance spectroscopy (EIS) data were provided in Figure 3c. However, it seems that there are some discrepancies in Ohmic resistance values between the samples. Similar Ohmic resistance values should be obtained under identical systematic conditions. Please confirm the experimental details during EIS.
4. It is revealed in Figure 3a that CFS-ACs/CNT catalyst shows the best OER activity among the samples, shown through LSV curves. However, when compared with FeS/CNT and CoS/CNT catalysts, the slope of the LSV curves differ (FeS/CNT and CoS/CNT catalysts show a steeper curve after 1.5 V

vs. RHE). It seems FeS/CNT and CoS/CNT catalysts will exhibit a better OER performance than CFS-ACs/CNT beyond the current density of 80 mA/cm² (y-axis limit in Figure 3a).

5. The authors propose the existence of 'CoFeS cluster' on the CNT support in Figure 1e. However, the domains for the respective Co, Fe and S don't seem to overlap with one another quite well, judging from the EDS mapping results provided. (Especially sulfur)

6. In page 8, it is described that 'Fe²⁺ XPS peak of CFS-ACs/CNT is shifted by approximately 2.13 eV'. XPS peaks corresponding to same species being shifted by more than 2 eV seems unrealistic. Perhaps it could imply the presence of other Fe species in different oxidation states? If not, at least a proper XPS peak assignment reference should be cited. Also, deconvoluted XPS peaks shown in Figure S5a is not well-defined.

7. XPS deconvolution results shown in Figure S7, depicting S 2p level is not very convincing as the M-S peak is completely submerged within the C-S-C peak. It would be better if there were other experimental results to confirm the existence of M-S species, or maybe provide additional explanation.

8. What is the exact role of sulfur? How does sulfur participate in enhancing the catalytic activity of OER? The authors assert that sulfur enhances the stability of the catalysts, but stability tests shown in Figure 3e compares CFS-ACs/CNT and FeS/CNT. (which both contains sulfur) Also, is the spin state of the catalysts altered by the presence of sulfur?

9. This work employed various techniques to determine the spin state of the catalysts. NEXAFS (near edge X-ray absorption fine structures) is an another tool that can be used to elucidate spin states. What are the authors' opinion on that?

10. Oxidation and spin state of Co and Fe species are susceptible to changes during OER. What about after the reactions? Also, what about the Co/Fe/S atomic compositions and distributions after the reactions? The authors showed that Fe²⁺ species are oxidized into Fe³⁺ after OER tests. Is this process reversible?

11. What is the content of Co, Fe, and S in CFS-ACs/CNT catalysts? Would altering the Co, Fe and S content change the OER catalytic activity? (for example, introducing higher amounts of Co, Fe and S onto CNT)

12. In Figure 4b, EXAFS data at different potentials are provided, up to 1.445 V. However, this potential corresponds to only about 20 mA/cm² over the CFS-ACs/CNT catalyst. What would happen at potentials higher than 1.445 V?

13. I may have misunderstood the authors' intentions, but the authors proposed an OPM (or DSSM) mechanism which does not necessarily involve OOH species. However, in-situ Raman data shown in Figure 4c reveals the presence of FeOOH species at high potentials. Could the authors provide further explanations? Also, what about the presence of CoOOH species?

14. It is said in the manuscript that catalytic activity was enhanced due to the increased number of active sites. Is there any possibility that 'intrinsic activity' of the individual active sites may have been enhanced as well?

15. Catalyst synthesis method shown in Figure 1a might be misleading. It seems like Co, Fe and S precursors were introduced altogether at the same time, like a one-pot synthesis.

16. There are some trivial grammar errors and typo throughout the manuscript.

Reviewer #3 (Remarks to the Author):

The paper reports synthesis of a well performing electrocatalyst that is composed of single domain ferromagnetic nanoclusters catalyst on carbon nanotube. Authors claim observing an unconventional, dual-site synergistic mechanism involving Co and Fe cations, which breaks the scaling relation that limits the performance of catalysts for oxygen evolution reaction (OER). The finding itself is definitely worth publication. However, provided discussion of the measured and computed data is sort of incoherent with plenty of results and scenarios discussed, without clear highlights. The claimed to be observed middle spin state configuration of Fe is rather unusual and would require comparison with other compounds showing similar spin state. So to say, extraordinary claims require very detailed and in-depth analysis and discussion, which unfortunately is not provided. Last, but not least, the applied simple DFT computational method is also questionable when applied to transition metal elements, especially for analysis of subtle spin arrangements.

Detailed comments:

- 1) Fe³⁺ MS should be discussed against similar spin configuration already observed in other compounds. It is not clear how such a configuration forms and if at all is permitted.
- 2) Introduction: scaling relation involves estimation of theoretical, thermodynamic overpotential, which is different from the measured kinetic overpotential. Please correct the discussion.
- 3) Introduction: "catalyst itself lattice" – remove lattice. In general, there are many grammar issues, which have negative impact on overall understanding of the manuscript. Profession proofreading is advised.
- 4) Introduction, end: when discussing spins of Fe(II), LS and MS, the reason for such spin configurations should be provided.
- 5) Introduction, last paragraph: are these results of own studies, the reported studies. It is not clear. Anyway, this part belongs rather to conclusion part, not to the introduction.
- 6) Results: "The more unpaired electrons can lead to a larger degree of d-p orbital overlap between the active site electron orbital filling and the oxygen intermediate, facilitating the adsorption/desorption of oxygen intermediates of the ferromagnetic catalyst" – it is not clear how this should happen.
- 7) Results: "In which, the main peak is fitted to one doublet of D1 (blue), assigned to M.S Fe(II)." - Please explain the fitting. Is Fe(II) MS common, ever observed? Besides, poor grammatical construction. In later text: "less than that those..." - remove "that".
- 8) Results: "Consistent with previously reported⁴⁷, Fe species were oxidized to a +3 valence state..." - please explain what exactly was reported previously. Later: "is gradually shifts" – poor grammar, correct.
- 9) Results: When reporting Fe-O bond lengths, please discuss which which spin state these correlate the best.
- 10) "the following DFT studies were conducted comparing it to the typical Co(OH*)-Fe(OOH*) and single-site Fe-OOH* pathway." - not understandable, reword. Besides, using just DFT to compute compounds with Co and Fe is not enough, especially when subtle things like spin arrangements are considered. Majority of studies apply DFT+U or hybrid functionals method.
- 11) Computations: It is not explained how different spin configurations were computed. Also authors

claim computing Gibbs free energies. How then the essential entropy term was estimated?

12) Overall, could be a nice paper, but if discussion is focused, key results in-depth discussed with providing convincing arguments supported by previous studies and if adequate computational approach were applied.

To Reviewer #1 :

The authors report an unconventional Co-Fe dual-site synergistic mechanism (DSSM) on a single-domain ferromagnetic catalyst CoFeS_x nanoclusters on carbon nanotube (CNT) catalyst (CFS-ACs/CNT), which can effectively break the scaling relationship without sacrificing stability. The authors claimed two different types of sites synergistically promote the production of Co-O-O-Fe intermediates, thereby benefiting accelerating the release of O₂. And various experiments and characterization have been conducted to support above conclusion. Therefore, I would like to recommend this manuscript to be accepted after the following considerations are fully addressed.

We appreciate you for giving us such a valuable opportunity to improve on the quality of our manuscript. Thank you for your careful review of the present work and the valuable comments on our manuscript. A list of point-to-point response was prepared as bellow. We sincerely hope that the revised manuscript can meet the requirement of your suggestions.

Detailed comments:

Q1. In Figure 1d, what is the specific crystal facet of lattice fringe in STEM image.

Answer: Thanks for the reviewer's valuable suggestions. As a transitional state from supported isolated metal atoms to metal nanoparticles, the nanoclusters require countable numbers of atoms in each cluster, and limited size range (normally smaller than 2 nm) (*Ding Ma, et. al., ACS Catal. 2020, 10, 11011-11045*). The metastable structure determines its weak crystallinity, so it is difficult to obtain obvious the specific crystal phase structure and exposed crystal faces (*J. Am. Chem. Soc. 2018, 140, 2812-2820; Proc. Natl. Acad. Sci. U. S. A. 2018, 115, 7700-7705*). As shown in Figure 1d, red circles showed the transitional state CoFeS_x nanoclusters (1.5~2 nm), and no specific crystal phase structure and exposed crystal faces are not observed. In addition, XRD pattern of CFS-ACs/CNT only exposed strong peaks belongs to carbon support with high graphitization, which implies CoFeS_x does not show a final metal nanoparticles form with good crystallinity. In this regard, we believed that CoFeS_x is stably anchored on the carbon carrier in the form of nanoclusters (1.5~2 nm).

Q2. Author conduct VSM and ⁵⁷Fe Mössbauer spectrum to confirm the spin state of Fe atoms, how about Co atoms and more experiment should be conducted to analyze the spin configuration of Co

atoms.

Answer: Thanks for the reviewer's valuable suggestions. To better evaluate the spin state of Fe and Co atom of CFS-ACs/CNT catalyst, we firstly accomplish temperature-dependent magnetic susceptibility ($M-T$) measurement to obtain the hybrid spin state (medium spin (M.S), $t_{2g}^5 e_g^1$) of CoFeS_x . Secondly, ^{57}Fe Mössbauer spectrum confirms the M.S. of Fe species ($t_{2g}^5 e_g^1$). Therefore, there are no excess unpaired electrons assigned to the Co site, and it is reasonable to assume that Co is stable in a low-spin state ($t_{2g}^6 e_g^0$).

Inspired by your suggestion, Co L-edge near edge X-ray absorption fine structures (NEXAFS) spectra is carried out to investigate the intensity of Co 3d empty/partially filled electronic state of CFS-ACs/CNT and CoS_x/CNT . As shown in **Figure R1(2f)**, two peaks of Co L_3 -edges are dominated by enhanced "white line" features and implies unoccupied t_{2g} and e_g states, respectively. Two main peaks appear in the energy region of Co L_3 -edge, denoted as A1 and A2. The peak A1 can be assigned to the transition from $2p_{3/2}$ to the $3d_{z^2}$ orbitals, while the peaks A2 is originated from the transition to $3d_{x^2-y^2}$ orbitals (*J. Chem. Phys.*, 2012, 137, 054306; *Nat. Commun.*, 2022, 13, 605). The unoccupied states of Co e_g orbitals of CFS-ACs/CNT and CoS_x/CNT samples can be directly reflected by the absorption intensity ratio of L_3/L_2 edge, it largely reflects the spin ground state given by the crystal field effect. CoS_x/CNT displays a larger L_3/L_2 intensity ratio of 3.5, while that of CFS-ACs/CNT is 2.5 (*Phys. Rev. B* 2009, 80, 014508; *Angew. Chem. Int. Ed.* 2023, 62, e202212335). Therefore, the suppression of the splitting energy of CoS_x/CNT suggests a weak crystal field effect that favors a higher e_g orbital occupancy of the Co center. Combined with $M-T$ curves, CoS_x/CNT catalyst exposes a unfilled e_g orbitals occupancy, called low spin state ($\text{Co}^{3+} t_{2g}^6 e_g^0$). No surprisingly, this is consistent with our initial reasonable assumptions.

Based on the results, the manuscript has been revised accordingly in **Page 11 line 21 to Page 12 line 14**.

Figure R1(2f). Co L-edge XANES spectra of CFS-ACs/CNT and CoS_x/CNT.

Q3. The current density normalized with the ECSA should also be provided to reveal what is major contribution on current density increase.

Answer: Thanks for the reviewer's valuable suggestions. According to the reviewer's comments, LSV curves revised by the current density normalized with the ECSA is provided in **Figure R2(S25)**.

Figure R2(S25). ECSA calibrated LSV curves of CFS-ACs/CNT, FeS_x/CNT, CoS_x/CNT and pure CNT.

The manuscript has been revised accordingly:

“The ECSA-corrected LSV curves (Figure S25) obtained after normalizing the current density using electrochemically active area (ECSA) show that CFS-ACs/CNT also has the highest intrinsic activity”. (Line 6-9 page 14)

Experiment: “The effective electrochemical active surface area (ECSA) can be obtained by the following equation: $ECSA = Cdl/Cs$. Cs is the specific capacitance in 1 M KOH ($Cs = 0.040$ mF) and Cdl is the electrochemical double layer capacitance measured using cyclic voltammetry curves (CV) at different scan rates”. (Line 1-3 page 23)

Q4. One major concern is the chemical stability in alkaline solution during OER process, whether the catalyst is transferred to other phase after oxygen electrocatalysis? In the current work, the author fails to discuss this matter as no structural characterization was conducted nor presented after the durability test after OER.

Answer: Thanks for the reviewer’s valuable suggestions. Yes, during OER process, especially in hydroxyl-rich environment, TM-based electrocatalysts inevitably undergo a certain degree of surface reconstruction process to form amorphous oxides/oxyhydroxides shell under the action of electrolyte hydrolysis and anode polarization, as referred results in *Nat. Commun.*, 2023, 14, 1949.

The detailed analysis are as follows:

Figure R3(S30). XPS spectra after long-term stability of about 633 h. (a) Co 2p, (b) Fe 2p, (c) S 1s and (d) O 1s.

➤ . As shown in the detailed XPS analysis after long-term stability in **Figure R3(S30)**, medium spin-state Fe²⁺ ($t_{2g}^5 e_g^1$) was transformed to Fe³⁺ ($t_{2g}^4 e_g^1$) (FeOOH species *etc.*). After constant >600 h OER test, the surface of CoFeS_x was partially oxidized to CoFe-based

oxides/oxyhydroxides. However, it should be pointed out that the spin occupation state of Fe^{3+} ($t_{2g}^4 e_g^1$) species remains unchanged, and the optimal oxygen intermediate adsorption are still maintained that are conducive to OER performance (*Nat. Commun.*, 2022, 13, 605). Based on the reviewer's comments, in addition to the XPS spectra analyzing the structural evolution after OER process, the phase transfer of CoFeS_x during the OER was also examined by using TEM images, *ex-situ* XRD techniques, and Raman spectrum, the details are as follows.

- As shown in **Figure R4(S29)**, TEM images after OER stability of CFS-ACs/CNT did not show significant particles produced by agglomeration of clusters.

Figure R4(S29). TEM images of CFS-ACs/CNT after OER stability test.

- As shown in **Figure R5(S29)**, XRD pattern after OER stability shows that three weak diffraction peaks belongs to Co(Fe)-OOH , and two weak peaks located at 26.6° (002) and 43.0° (100) belongs to carbon nanotube. As explained in the response for the comment 1 and TEM images, the clusters do not agglomerate into a highly crystalline phase after stabilization, thus XRD only obtained weak characteristic peaks of FeOOH and graphitic carbon.

Figure R5(S29) XRD pattern of CFS-ACs/CNT after OER stability test.

- As shown in **Figure R6(S29)**, Raman spectroscopy after OER stability show that the main peak at 216 and 472 cm^{-1} are attributed to E_g and A_g modes of Fe-O and $\text{Co}_x\text{Fe}_y\text{-O}$ bonds, respectively (*ACS Catal.* 2022, 12, 3743-3751; *J. Electro. Soc.*, 2017, 164 (9) H621-H627). The existence of CoFeOOH phase shows that the partially oxidized on the surface of CoFeS_x , which is matched with XPS spectra.

Figure R6 (S29) Raman spectroscopy of CFS-ACs/CNT after OER stability test.

Overall, the results show that after long-term stability testing, irreversible surface remodeling occurs on the surface of the catalyst to form metal oxides/oxyhydroxides species. This is consistent with the results of *in-situ* Raman analysis. Based on the results, the

manuscript has been revised and added accordingly in **Supplementary Note 1 and 2 of Supplementary Materials**. The manuscript has been revised accordingly in **line 7-14 page 15**.

Q5. The authors need to provide the fitting model for the calculation of the coordination number in EXAFS fitting.

Answer: Thanks for the reviewer's valuable suggestions. We apologize for this oversight and were reminded by the reviewer to validate the fitted catalyst structure.

Figure R7(S14) Comparison between the experimental Fe K-edge XANES spectra of CFS-ACs/CNT catalyst and the theoretical spectra calculated based on CoFeS₄ structure embedded in the carbon matrix.

According to Table S1, CoFeS₄ structure (illustrations inside **Figure R7 (S13)**) is corresponded with the coordination number of Fe-S (~5) and Co-S (~6). For the further verify the structure, Fe K-edge XANES spectra of theoretical spectra calculated based on CoFeS₄ structure was obtained. As shown in **Figure R7(S13)**, all the spectra feature of Fe K-edge XANES can be well produced using CoFeS₄ structure.

Q6. The equivalent electrical circuit to fit the EIS should be also provided.

Answer: Thanks for the reviewer's valuable suggestions. According to the reviewer's comments, the equivalent circuit diagram has been added into **Figure R8 (Fig. 3c)**.

Figure R8 (Fig. 3c) EIS spectra of CFS-ACs/CNT, FeS_x/CNT, CoS_x/CNT and pure CNT.

To Reviewer #2:

The paper describes studies of Co-Fe dual metal-based catalysts supported on sulfur modified carbon nanotubes for OER in alkaline solution. This field of research has received much attention in the past and represents an important system for understanding the mechanism of the OER reaction. The authors describe samples prepared by thermo-chemical treatment of carbon nanotubes while also proposing an OER pathway based on oxide path mechanism (OPM). Quite interesting results were obtained through diverse characterization techniques. However, some of the points proposed by the authors are not very convincing and/or misleading. Many points in this work should be addressed, as presented in detail below, for publication in Nature Communications to be justified.

We appreciate you for giving us such a valuable opportunity to improve on the quality of our manuscript. Thank you for your careful review of the present work and the valuable comments on our manuscript. A list of point-to-point response was prepared as bellow. We sincerely hope that the revised manuscript can meet the requirement of your suggestions.

Detailed comments:

Q1. There are diverse kinds of ferromagnetic catalysts other than the combination of cobalt and iron. Is there a specific reason for choosing cobalt and iron as active phase metals in this work?

Answer: Thanks for the reviewer's thoughtful suggestion. As pointed out by Gracia *et al.*, to improve the OER efficiency, it is necessary to consider spin selected electron transfer in order to facilitate the generation of triplet dioxygen, which is thermodynamically favorable over the singlet counterparts. One of the theoretical discoveries given by Gracia is that catalysts with ferromagnetic (FM) properties may facilitate the spin-selective electron transfer from singlet reactants toward triplet dioxygen. (*Angew. Chem. Int. Ed.* 2023, e202301721)

We will explain your question with four points:

- We introduce the main ferromagnetic elements to construct ferromagnetic catalysts. Ferromagnetic substances refer to the additional magnetic field that produces the same strong additional magnetic field as the external magnetic field under the action of the magnetic field, which can be understood as greatly enhancing the original magnetic field (“*Novel*

electromagnetic nondestructive testing technology” edited by Huang Songling, Tsinghua University). Meanwhile, there are two main bases for determining whether or not they are ferromagnetic: a) the presence of a non-fully occupied electronic state in the electron shell layer of an atom is necessary to produce ferromagnetism; b) heisenberg theory: for transition metals, the 3*d* states of atoms do not differ much in energy from the 4*s* states, so their electron clouds will overlap, causing redistribution of electrons between the *s* and *d* states, resulting in the electron exchange energy (E_{ij}). The essence of this electron exchange effect of neighboring atoms is still the electrostatic force forcing the electron spin magnetic moments to be aligned in parallel, acting as if it were a strong magnetic field.

So far, **only four metal elements have been ferromagnetic above room temperature, namely iron (Fe), cobalt (Co), nickel (Ni) and gadolinium (Gd)**. Gd, as rare earth metals, exposes the unfilled 4*f* or 5*d* electron shell, and there is an unoffset spin moment, so their paramagnetism is strong, therefore the magnetic susceptibility is large, obeying the Curie-Weiss law. Due to the unfilled 3*d* to 5*d* electron shell layers of the transition metal elements, the uncounteracted spin magnetic moments of the electrons form the intrinsic magnetic moments of the crystalline ions, resulting in strong paramagnetism.

- Therefore, the choice of two ferromagnetic substances for the construction of FM catalysts ensures to some extent the maintenance of ferromagnetism after complex synthesis processes (As for the bimetallic sites exhibiting faster OER kinetics than single-site catalysts, and the complex and unexplained ternary and higher catalytic mechanisms are not described here). Secondly, from the application point of view, that is, to promote OER performance. A large number of non-precious metal catalysts have been reported for OER electrocatalysis in alkaline conditions, including mainly the Co-based, Ni-based, and Fe-based materials. Among them, Co-Fe bimetallic ferromagnetic catalysts show a leading edge with high OER activity in alkaline environments (*Nat. Commun.*, 2023, 14, 4791; *Energy Environ. Sci.*, 2023, 16, 1685-1696; *Adv. Mater.*, 2017, 29, 1701546). Qiu *et. al.* screened CoFe heteronuclear diatomic catalysts from CoFe, CoNi, FeNi by DFT calculations in theoretically superior OER activity (*Nano Lett.*, 2022, 22, 8, 3392-3399).
- The spin state affects the OER activity through modulating the e_g occupancy of active species, which is strongly affects the binding of oxygen intermediates. Co with higher valence states

(+3 or even higher) by losing unpaired electrons shows optimal affinity for surface oxygen. The medium spin (M.S., $t_{2g}^5 e_g^1$) of Fe sites (Fe^{2+}) has appropriate e_g occupancy, which is expected to confer excellent OER performance.

- This work is inspired by a recent report of spin-polarized OER process of Prof. Zhichuan J. Xu (*Nat. Commun.*, 2021, 12, 2608; *Angew. Chem. Int. Ed.*, 2023, e202301721). The first step of the electron transfer process in the OER by the action of spin-polarized electrons using ferromagnetic CoFe_2O_4 . The generation of triplet state O_2 is dominated by the exchange effect of spin-polarized electrons (*Nat. Commun.*, 2021, 12, 2608). In addition to using an external magnetic field to tune the magnetic domain structure of FM catalysts to induce spin-polarized water oxidation, the construction of single-domain ferromagnetic-type catalysts (small-size CoFe_2O_4 nanoparticles), which takes advantage of intrinsic spin ordering, can similarly accelerate the release of oxygen from the ternary state (*Angew. Chem. Int. Ed.*, 2023, e202301721), and we believe that it's an interesting and novel perspective.

Q2. In EXAFS analysis (Figure 2b and 4b, for example), there should be no distinctive peaks in the range of < 1 angstrom in R space. Perhaps the authors should reconsider the EXAFS fitting conditions.

Answer: Thanks for the reviewer's thoughtful suggestion. We recheck and reprocess the data with a technical professor. The absorption spectrum of the K-edge of Fe, i.e., the synchrotron fine diffraction absorption spectrum of the Fe element near 7112 eV, is processed using the Fe foil as reference. Subsequently, the Fourier transform was performed to obtain the treated XANES spectra (**Figure R1 (Fig. 2b, Fig. S11, Fig. 4b)**), which show that the shorten Fe-S/Co-S bonds of CFS-ACs/CNT. The shorten of atom-atom distance is in line with the conditions for the formation of a new O-O coupling mechanism proposed later (*Nat. Commoun.* 2021, 4, 1012-1023). We also refer to some similar works, and our data are consistent with those in the literature (**Figure R2**).

Figure. R1 (Fig. 2b, Fig. S11, Fig. 4b). (a, b) Fourier-transform EXAFS spectra of CFS-ACs/CNT, FeS_x/CNT and CoS_x/CNT; (c) Fe K-edge FT-EXAFS during OER process with the different potentials (vs. RHE) of OCV, 1.245 V, and 1.445 V in 1 M KOH, respectively.

Figure. R2. (a) FeS₁N₃ (*Angew. Chem. Int. Ed.* 2021, 60, 25296-253); (b-c) Fe_xCo_yS (*Adv. Energy Mater.* 2022, 12, 2201608).

Q3. Electrochemical impedance spectroscopy (EIS) data were provided in Figure 3c. However, it seems that there are some discrepancies in Ohmic resistance values between the samples. Similar Ohmic resistance values should be obtained under identical systematic conditions. Please confirm the experimental details during EIS.

Answer: Thanks for the reviewer's valuable suggestions. Firstly, it should be pointed out that EIS analysis was recorded at 1.445 V (vs. RHE). For maintaining the uniformity of the electrolyte environment, freshly formulated electrolytes (1.0 M KOH) were used for the EIS testing of different catalyst. Therefore, considering the reviewer's suggestion, electrochemical impedance spectroscopy (EIS) data were obtained again with the above condition (**Figure R3(Fig.3c)**):

Figure R3(Fig.3c). EIS spectra of CFS-ACs/CNT, FeS_x/CNT, CoS_x/CNT and pure CNT.

In the same test environment, the solution resistances (R_s) of CFS-ACs/CNT, FeS_x/CNT and CoS_x/CNT catalysts are nearly identical, while CFS-ACs/CNT shows lower charge transfer resistance (R_{ct}) of 4 Ω (*Joule* 2021, 5, 2164-2176; *J. Am. Chem. Soc.* 2022, 144, 8204–8213), which is mainly due to the difference in conductivity caused by the different materials.

Q4. It is revealed in Figure 3a that CFS-ACs/CNT catalyst shows the best OER activity among the samples, shown through LSV curves. However, when compared with FeS/CNT and CoS/CNT catalysts, the slope of the LSV curves differ (FeS/CNT and CoS/CNT catalysts show a steeper curve after 1.5 V vs. RHE). It seems FeS/CNT and CoS/CNT catalysts will exhibit a better OER performance than CFS-ACs/CNT beyond the current density of 80 mA/cm² (y-axis limit in Figure 3a).

Answer: Thanks for the reviewer’s valuable suggestions. Inspired by the reviewer, we performed a methodological check of the LSV curves with iR compensation for all prepared catalysts of Fig. 3a. FeS_x/CNT and CoS_x/CNT comparison samples surprisingly show smaller overpotentials in the high current density range, which is inexplicable and contrary to our repeated test results. The detailed analysis are described as follows:

- Examining the LSV curves carefully, we found a data processing error, the FeS_x/CNT and CoS_x/CNT catalyst data were iR compensated, while the other data were not compensated. For further validation we compared the curves before and after the compensation and added the compensated curves to the **Supplementary Fig. S21**, as shown in the **Figure R4**. Meanwhile, $i-t$ curves (**Figure R4f**) of CFS-ACs/CNT, FeS_x/CNT and CoS_x/CNT at 100 mA cm⁻² with 80% iR compensation were supplemented. As shown in Figure R4a-e, at the low current density of 20 mA cm⁻², the overpotentials of the LSV curves before and after iR compensation do not

differ significantly from each other. However, at the high current density of 100 mA cm^{-2} , iR compensation is very necessary, the overpotential is significantly reduced. Therefore, to better evaluate the catalyst stability, i - t curves with 80% iR compensation were examined at 100 mA cm^{-2} . Similarly, the CFS-ACs/CNT catalyst exhibits excellent stability at 100 mA cm^{-2} , there are no significant drop of current density under continuous stability testing of 70 h. The manuscript has been revised accordingly in **line 10-13 page 13** and **line 18-22 page 14**.

Figure R4(S21) (a-d) LSV curves before and after iR compensation of CFS-ACs/CNT, FeS_x/CNT, CoS_x/CNT and pure CNT. (e) LSV curves of CFS-ACs/CNT, FeS_x/CNT, CoS_x/CNT and pure CNT with 80% iR compensation. (f) The chronoamperometry curve of CFS-ACs/CNT, FeS_x/CNT and CoS_x/CNT at 100 mA cm^{-2} with 80% iR compensation.

Figure R5(S19) (a) LSV curves and (b) Tafel slopes of CFS-ACs/CNT, FeS_x/CNT, CoS_x/CNT, pure CNT, commercial IrO₂ on Ni foam substrate **without iR compensation**.

➤ The curves in **Fig. 3a** were changed to LSV curves without iR compensation. (Considering the characteristics of iR compensation, at low current densities, there is little difference in iR compensation correction, so we still maintain the original nature of the previous data and use the uncompensated data as a reference) Meanwhile, the associated Tafel slope was recalculated and corrected as shown in **Figure R5(S19)**:

Based on the results above, the manuscript has been revised accordingly in **Fig. 4a** and **Figure S19**.

Q5. The authors propose the existence of ‘CoFeS cluster’ on the CNT support in Figure 1e. However, the domains for the respective Co, Fe and S don’t seem to overlap with one another quite well, judging from the EDS mapping results provided. (Especially sulfur)

Answer: Thanks for the reviewer’s valuable suggestions. We will respond to your questions in the following aspects:

1) Firstly, we have to point out that the relatively low atomic number of the S element leads to its different contrasts between Co(Fe) and S atom (*Nat. Catal.*, 2022, 5, 503-512; *Nat. Commoun.*, 2020, 11, 5075). Heavy element Co/Fe is highly lined, reflecting relatively bright images. It is also evident from the EDS maps that the FeCo intensities and brightnesses are extremely overlapping.

Figure R6 (a) EDS images of FeCoO_x-Vo-S (*Angew. Chem. Int. Ed.* 2020, 59, 14664); (b) Atomic HAADF-STEM image of (NiCo)S_{1.33} (*Nat. Commoun.*, 2023, 14, 1949); (c) Corresponding elemental mappings of Ni(Fe)(OH)₂-FeS_x (*Nat. Commoun.*, 2020, 11, 5075).

2) Secondly, considering the local in homogeneous characterization on catalyst surface during the TEM mapping test, we've done multiple iterations of this test. As shown in the figure below:

Figure R8 (Fig. 1e). (a-e) Energy dispersive X-ray spectroscopy (EDS) mappings of individual elements (C, S, Fe, and Co).

The domains for the respective Co, Fe and S well-overlap with one another quite well, judging from the EDS mapping the above figure. And it is reasonable that the brightness of the S element is relatively stronger due to the S admixture to the surface of the carbon nanotubes, which coincides with the atomic ratio of Co/Fe/S obtained by XPS spectra in Table S3. **The diagrams (Fig. 1e)** in the manuscript have been replaced accordingly.

Q6. In page 8, it is described that ‘Fe²⁺ XPS peak of CFS-ACs/CNT is shifted by approximately 2.13 eV’. XPS peaks corresponding to same species being shifted by more than 2 eV seems unrealistic. Perhaps it could imply the presence of other Fe species in different oxidation states? If not, at least a proper XPS peak assignment reference should be cited. Also, deconvoluted XPS peaks shown in Figure S5a is not well-defined.

Answer: Thanks for the reviewer’s valuable suggestions. We have reviewed a large amount of literatures and found that as the reviewer stated this is anomalous for XPS peaks corresponding to same species being shifted by more than 2 eV. Therefore, we re-analyze the XPS data. We examined the source data for both samples of CFO-p/CNT and CFS-ACs/CNT and found a major error: the calibration was performed against C1s, with an XPS standard C1s of 284.8 eV binding energy, and a calibration charge deviation of $\Delta E = 284.8\text{eV} - 284.668\text{eV} = 0.132\text{eV}$ was derived by comparing the experimental C1s binding energy of 284.668 eV. The diagram after charge correction is shown below **Figure R9 (S5b)**.

Figure R9 (S5) XPS spectra. Co 2p (a) and Fe 2p (b) of CFO-p/CNT and CFS-ACs/CNT, respectively.

As shown in **Figure R9 (S5)**, the Co 2p XPS spectra were re-attributed and analyzed in detail as follows: the Co 2p XPS spectra of CFO-p/CNT and CFS-ACs/CNT displayed two major Co 2p_{3/2} and Co 2p_{1/2} spin-orbit splitting. The Co 2p peak for CFS-ACs/CNT located at 779.9 (782.4 eV) and 794.9 eV (798.0 eV) can be assigned to be Co³⁺ and Co²⁺, respectively. A more positive peak

shift is further observed in CFS-ACs/CNT, suggesting that the Co species are partly able to form a higher valence state (+3).

Finally, the analysis about **Figure S5**, we have also revised and highlighted accordingly in the manuscript **in line 5-9 page 14**.

Q7. XPS deconvolution results shown in Figure S7, depicting S 2p level is not very convincing as the M-S peak is completely submerged within the C-S-C peak. It would be better if there were other experimental results to confirm the existence of M-S species, or maybe provide additional explanation.

Answer: Thanks for the reviewer's valuable suggestions. The S 2p spectrum with the characteristic peak situated at 161~164 eV was corresponding to sulfur species in C-S-C peak and M-S units (*Angew. Chem. Int. Ed.*, 2020, 59, 14664; *Adv. Energy Mater.* 2022, 2103275). Therefore, XPS spectra are difficult to fully distinguish between C-S-C and M-S peak and resulting in the appearance of M-S peak is completely submerged within the C-S-C peak.

Figure R10 (S7) (a) XPS spectra for S 2p of CFS-ACs/CNT; (b) Raman spectrum of CFS-ACs/CNT, FeS_x/CNT and CFO-p/CNT.

Following the suggestion of the reviewers, the formation of M-S was further verified by Raman spectroscopy. Comparison with the CoFe-precursor, as shown in **Figure R10 (S7)**, the peak at 322, 340 and 375 cm^{-1} are attributed to T_g , E_g and A_g modes of Fe-S and Co(Fe)-S bonds, respectively (*ACS Catal.* 2022, 12, 3743–3751; *Chem. Eng. J.*, 2020, 379, 122240). Based on the results, the manuscript has been revised accordingly in **line 13-17 page 8**.

“The XPS spectrum of S 2p is divided into three peaks: Fe(Co)-S at 162.54 eV, C-S-C coordination at 163.74 eV, and oxidation state S-O at 168.98 eV (S 2p_{3/2}) and 170.24 eV (S 2p_{1/2})” was changed to “*The XPS spectrum of S 2p (Figure S7a) is divided into three peaks: Fe(Co)-S or C-S-C coordination at 162.5 eV, and oxidation state S-O at 168.98 eV (S 2p_{3/2}) and 170.24 eV (S 2p_{1/2}). As shown in Figure S7b, Comparison with the CoFe-precursor, the peak at 322, 340 and 375 cm^{-1} are attributed to T_g , E_g and A_g modes of Fe-S and Co(Fe)-S bonds.*”

Q8. What is the exact role of sulfur? How does sulfur participate in enhancing the catalytic activity of OER? The authors assert that sulfur enhances the stability of the catalysts, but stability tests shown in Figure 3e compares CFS-ACs/CNT and FeS/CNT. (which both contains sulfur) Also, is the spin state of the catalysts altered by the presence of sulfur?

Answer: Thanks for the reviewer’s valuable suggestions.

(1) What is the exact role of sulfur?

During the synthetic processes, the decomposition of thioacetamide during the hydrothermal reaction not only provides S sources for the formation of a CoFeS_x but also creates the desired ‘operando acidic environment’ to facilitate the synthesis of atomically dispersed CoFeS_x nanocluster by suppressing the formation of metallic nanoparticles (*Nat. Nanotechnol.*, 2023, 18, 763-771). In other words, the introduction of S atom successful bolting of CoFe active sites onto CNT and further avoidance of possible agglomeration phenomena. As illustration in **Figure R11**, TEM images of CoFeO_x/CNT showed that a large number of sheet catalysts (pure CoFeO nanosheets) were dispersed on CNTs.

Figure R11(S4) TEM images of the CFO-p/CNT (yellow circle is CFO-p nanosheets precursor).

(2) *How does sulfur participate in enhancing the catalytic activity of OER?*

Answer: As discussed above, sulfur not only can provide sulfur source to synthesize CoFeS_x compound, but can control the size of CoFeS_x to prepare clusters. Well-known, sulfides have long been regarded as a high-performance electrocatalyst for oxygen evolution due to their satisfactory OER activity and acceptable stability in appearance. However, the thermodynamic instability of sulfides against oxidative working OER conditions has long perplexed the researchers (*Energy Environ. Sci.*, 2022, 15, 3257-3264). In this manuscript, the introduction of sulfur successfully constructs CoFeS_x clusters, which can not only enhance OER performance due to the optimal intrinsic activity of sulfides, but can anchor metal sites more stably to achieve strong long-term stability, which is a puzzle that has also long perplexed the researchers. As ‘Introduce’ section, compared to oxides, sulfides show a better conductivity or metallic property, where a larger atomic radius and smaller ionization energy of sulfur atoms contribute to a faster charge transfer (*Energy Environ. Sci.*, 2021, 14, 365-373; *J. Am. Chem. Soc.*, 2019, 141, 7005 – 7013). Hence, it would render the metal atom surface more electrophilic and lead to an increased affinity for (oxy)hydroxyl species, and induce the improved OER kinetics.

(3) *The authors assert that sulfur enhances the stability of the catalysts, but stability tests shown in Figure 3e compares CFS-ACs/CNT and FeS/CNT. (which both contains sulfur) Also, is the spin state of the catalysts altered by the presence of sulfur?*

Answer: Considering the electronic modulation induced by the change in catalyst structure, we further examine the changes in catalyst properties, especially stability. As shown in **Figure R12a, b**, we examined the magnetic properties of the CFO-p/CNT catalyst and the electronic spin state distribution by means of VSM test. Compared with CFS-ACs/CNT, CFO-p/CNT catalyst shows

poor saturation magnetization (M_s). Not only that, temperature-dependent magnetic susceptibility ($M-T$) measurement derives the effective magnetic moment (<1), which means fewer, or even close to zero, unpaired electrons. Therefore, the alteration of the electronic structure of the catalyst on the electron spin occupied states was not accidental. Subsequently, the OER activity and stability of CFO-p/CNT sample were assessed in **Figure R12c, d**. Especially the stability, as mentioned above, the introduction of S not only improves the electronic structure of the catalyst but also successfully attenuates the aggregation state of CoFeS_x clusters during the catalytic process by bolting on the metal sites to maintain its lasting stability. Therefore, as shown in **Figure R12d**, the CFO-p/CNT catalyst exhibits weak stability, with a 40% drop in current density after constant 70 h OER testing. Therefore, we apologize for the misunderstanding due to our inappropriate expression. What we meant to say about this sentence is that the introduction of S atoms can reduce the tendency of agglomeration of the clusters through the binding metal sites, thus enhancing the structural stability. We have fixed this inappropriate description in **line 22 page 8**, and supplement the stability data in **line 21-22 page 14**.

Figure R12 (S22b, S28) (a) Magnetic hysteresis loops of CFO-p/CNT at room temperature (300 K). (b) $M-T$ susceptibility χ and reciprocal $1/\chi$ of CFO-p/CNT. (c) LSV curve, and (d) $i-t$ curve of CFO-p/CNT.

Finally, TEM images (**Figure R13**) after 633 h stability of CFS-ACs/CNT did not show significant particles produced by agglomeration of clusters, which is consistent with the above

observations.

Figure R13 TEM images of CFS-ACs/CNT after OER stability.

Q9. This work employed various techniques to determine the spin state of the catalysts. NEXAFS (near edge X-ray absorption fine structures) is another tool that can be used to elucidate spin states. What are the authors' opinion on that?

Answer: Thanks for the reviewer's valuable suggestions. Inspired by the reviewers, we reviewed a many literature (*Nat. Commun.*, 2022, 13, 605; *Angew. Chem. Int. Ed.* 2023, 62, e202301075; *Angew. Chem. Int. Ed.* 2021, 60, 25296-25301; *Angew. Chem. Int. Ed.* 2023, 62, e202212335; *J. Chem. Phys.*, 2012, 137, 054306; *Phys. Rev. B* 2009, 80, 014508; *Adv. Mater.* 2022, 34, 2202240) reporting characterization techniques for verifying the evolution of spin states. There is no doubt that NEXAFS (near edge X-ray absorption fine structures) is an effective tool that can be used to elucidate spin states of single metal atom. Herein, we have added the Co L-edge NEXAFS spectra in the revised version.

In the "Results" section, to better evaluate the spin state of Fe and Co atom of CFS-ACs/CNT catalyst, firstly, we accomplished temperature-dependent magnetic susceptibility (M-T) measurement to obtain the hybrid spin state (M.S $t_{2g}^5 e_g^1$) of CoFeS_x . Secondly, ^{57}Fe Mössbauer spectrum confirms the M.S of Fe species ($t_{2g}^5 e_g^1$). Therefore, there are no excess unpaired electrons assigned to the Co site, and it is reasonable to assume that Co is stable in a low-spin state ($t_{2g}^6 e_g^0$). Co L-edge near edge X-ray absorption fine structures (NEXAFS) spectra can investigate the intensity of Co 3d empty/partially filled electronic state of CFS-ACs/CNT and CoS_x/CNT . As shown

in **Figure R1(2f)**, two peaks of Co L₃-edges are dominated by enhanced “white line” features and implied unoccupied t_{2g} and e_g states, respectively. Two main peaks appear in the energy region of Co L₃-edge, denoted as A1 and A2. The peak A1 can be assigned to the transition from $2p_{3/2}$ to the $3d_{z^2}$ orbitals, while the peaks A2 is originated from the transition to $3d_{x^2-y^2}$ orbitals (*J. Chem. Phys.*, 2012, 137, 054306; *Nat. Commun.*, 2022, 13, 605). The unoccupied states of Co e_g orbitals of CFS-ACs/CNT and CoS_x/CNT samples can be directly reflected by the absorption intensity ratio of L₃/L₂ edge, it largely reflects the spin ground state given by the crystal field effect. CoS_x/CNT displays a larger L₃/L₂ intensity ratio of 3.5, while that of CFS-ACs/CNT is 2.5 (*Phys. Rev. B* 2009, 80, 014508; *Angew. Chem. Int. Ed.* 2023, 62, e202212335). Therefore, the suppression of the splitting energy of CoS_x/CNT suggests a weak crystal field effect that favors a higher e_g orbital occupancy of the Co center. Combined with M-T curves, CoS_x/CNT catalyst exposes a low e_g orbitals occupancy, called low spin state ($\text{Co}^{3+} t_{2g}^6 e_g^0$). Not surprisingly, this is consistent with our initial reasonable assumptions.

Based on the results, the manuscript has been revised accordingly in **line 21 page 11 to line 14 Page 12**.

Figure R14(Fig. 2f) Co L-edge XANES spectra of CFS-ACs/CNT and CoS_x/CNT.

Not only that, the spin state of Fe and Co atom from CFS-ACs/CNT after OER stability also examined through NEXAFS spectrum. Fortunately, this further proves our inference, and lay a solid experimental foundation for the DSSM mechanism. The spin state after stabilization will be analyzed in detail in comment 10.

Q10. Oxidation and spin state of Co and Fe species are susceptible to changes during OER. What about after the reactions? Also, what about the Co/Fe/S atomic compositions and distributions after the reactions? The authors showed that Fe²⁺ species are oxidized into Fe³⁺ after OER tests. Is this process reversible?

Answer: Thanks for the reviewer's valuable suggestions. We will respond to each of these questions:

Q10-1. Oxidation and spin state of Co and Fe species are susceptible to changes during OER. What about after the reactions?

Answer: Oxidation state of Co and Fe species after the OER reactions was evaluated by XRD, XPS and Raman spectroscopy. In addition, XPS spectra to analyzing the structural evolution after OER process, the phase transition of CoFeS_x during the OER was also examined by using *ex-situ* XRD techniques and Raman spectrum.

Figure R15(S30) XPS spectra after long-term stability of about 633 h. (a) Co 2p, (b) Fe 2p, (c) S 1s and (d) O 1s.

- As shown in **Figure R16**, TEM images after OER stability of CFS-ACs/CNT did not show significant particles produced by agglomeration of clusters.

Figure R16 (S29). TEM images of CFS-ACs/CNT after OER stability

- As shown in **Figure R17**, XRD pattern after OER stability showed that three weak diffraction peaks belonging to Co(Fe)-OOH, and two weak peaks located at 26.6° (002) and 43.0° (100) belongs to carbon nanotube. The metastable structure determines its weak crystallinity, so it is difficult to obtain obvious the specific crystal phase structure and exposed crystal faces (*J. Am. Chem. Soc.* 2018, 140, 2812–2820; *Proc. Natl. Acad. Sci. U. S. A.* 2018, 115, 7700–7705), the clusters do not agglomerate into a highly crystalline phase after stabilization, thus XRD only obtained weak characteristic peaks of FeOOH and graphitic carbon.

Figure R17(S29) XRD pattern of CFS-ACs/CNT after OER stability.

- As shown in **Figure R18**, Raman spectroscopy after OER stability show that the main peak at 216 and 472 cm^{-1} are attributed to E_g and A_g modes of Fe-O and $\text{Co}_x\text{Fe}_y\text{-O}$ bonds, respectively (*ACS Catal.* 2022, 12, 3743–3751; *J. Electro. Soc.*, 2017, 164 (9) H621-H627). The existence

of CoFeOOH phase show that the partially oxidized on the surface of CoFeS_x, which is matched with XPS spectra.

Figure R18(S29) Raman spectroscopy after OER stability of CFS-ACs/CNT.

- Therefore, it is necessary to study the spin state of the CFS-ACs/CNT catalyst after stabilization. As shown in **Figure R19**, Fe,Co L₂, L₃-edge XANES spectra to investigate the density of Fe,Co 3d empty/partially filled electronic state after OER stability. As shown in **Figure R19a(Fig.4d)**, two peaks of Fe L₃ edges are dominated by enhanced “white line” features and implied unoccupied *t_{2g}* and *e_g* states, respectively. Therefore, the unoccupied states of Fe *e_g* orbitals of CFS-ACs/CNT samples can be directly reflected by the absorption intensity ratio of L₃ to L₂ edge, it largely reflects the spin ground state given by the crystal field effect (*Angew. Chem. Int. Ed.* 2023, 62, e202212335; *Phys. Rev. B* 2009, 80, 014508). Besides, CFS-ACs/CNT displays a larger L₃/L₂ intensity ratio of 1.6, while that of CFS-ACs/CNT after stability is 1.3. Therefore, the suppression of the splitting XANES structure of CFS-ACs/CNT suggests a weak crystal field effect that favors a higher *e_g* orbital occupancy of the Fe center. Fe²⁺ site of CFS-ACs/CNT catalyst is exposed a half-filled *d_{z2}* orbitals occupancy with the medium spin state (*t_{2g}⁵e_g¹*). Therefore, Fe³⁺ site after OER stability shows a lower *e_g* orbitals occupancy closing to 1, due to the approximate intensity ratio before and after stability. Similarly, as shown in Co L-edge XANES (**Figure R19b(S32)**), CFS-ACs/CNT displays a smaller L₃/L₂ intensity ratio than that of CFS-ACs/CNT after stability. Co³⁺ site of CFS-ACs/CNT catalyst exposes unfilled *e_g* orbitals with the low spin state (*t_{2g}⁶e_g⁰*). Therefore, Co³⁺ site after stability may be partially filled *e_g* orbitals with the medium spin state (*t_{2g}⁵e_g¹*) or high

spin state ($t_{2g}^4 e_g^2$). However, the evolution from L.S to H.S requires much lower splitting energy than pairwise energy (from unfilled to half-filled e_g occupancy), so the M.S is more compatible (*Angew. Chem. Int. Ed.* 2023, 62, e202216837).

Figure R19 (Fig.4d, S32). Fe, Co L-edge XANES spectra before and after stability of CFS-ACs/CNT.

Q10-2. Also, what about the Co/Fe/S atomic compositions and distributions after the reactions?

Answer: The surface compositions are investigated by XPS. According to the XPS spectra after OER stability, the surface of CoFeS_x was partially oxidized to CoFe(OOH) . Therefore, after stabilization, the O atom content increases with the decrease of active species Fe/Co. As shown in Table S3, Co/Fe/S atomic ratio (0.77%/0.64%/14.8%) after the reactions is close before the reactions (1.31%/1.28%/19.01%). The synchronous loss of the metal site means that the Co-Fe double active site is justified. Not only that, the intensity of the S 2p peak for CFS-ACs/CNT only shows slight degradation with a marginal decrease of S content from 19.01 to 14.8 at % (Table 3). Sulfur loss is feeble, which further demonstrate the structural stability after the introduction Co, S atom.

Table S3 Atomic distribution of CFS-ACs/CNT before and after OER stability.

Elements	Atomic% after OER stability for 633 h	Atomic% before OER stability
C	76.66	78.31
Co	0.77	1.31
Fe	0.64	1.28
S	14.8	19.01
O	7.13	0.09

Q10-3. The authors showed that Fe^{2+} species are oxidized into Fe^{3+} after OER tests. Is this process reversible?

Answer: This process is unreversible at the same condition. Well-known, during strongly oxidizing, high potential tests, the surface of the metal catalyst is often accompanied the dissolution and reconstitution to generate metal oxides (hydroxides). However, due to the widely observed sulphur atom loss during the sulphide's transform into a corresponding active oxide/hydroxide electrocatalyst, the TM sulphides showcase a more complicated restructuring mechanism compared to the pure oxides and hydroxides (*Nat. Catal.*, 2021, 4, 212-222). Driven by the applied voltage and rich hydroxyl molecules in solution, the sulphur atoms on the surface of $CoFeS_x$ clusters are prone to be partially substituted by oxygen and further induce the formation of an oxygen-sulphur coexisting $CoFeS(O)$ phase (*Nat. Commoun.*, 2023, 14, 1949). Therefore, it is inevitably form a sulphide/oxide core/shell structure. Surface reconstruction, lattice oxygen dissolution, and electrode surface redeposition are considered to be the main factors affecting catalyst stability. Investigate additional ways to control the degree of reconstruction so that the process is terminated at the desired point to stabilize the catalyst is a long-term research topic for the industrial development of OER catalysts (*Joule*, 2021, 5, 1704-1731; *Joule*, 2021, 18, 2164-2176). However, the present work maintains the balance between OER catalyst activity and stability from the perspective of catalytic mechanism mainly through the special O-O coupling pathway (DSSM), i.e., not only spanning the linear scaling relationship between $*OOH$ and $*OH$, but avoiding the structural collapse caused by the lattice oxygen migration and solvation during the operation of LOM mechanism.

Q11. What is the content of Co, Fe, and S in CFS-ACs/CNT catalysts? Would altering the Co, Fe and S content change the OER catalytic activity? (for example, introducing higher amounts of Co, Fe and S onto CNT)

Answer: Thanks for the reviewer's valuable suggestions.

- The Co (17.63 wt%_{ICP-OES}) and Fe (19.37wt%_{ICP-OES}) content of CFS-ACs/CNT is acquired by inductively coupled plasma-optical emission spectroscopy (ICP-OES), which was used as a standard for subsequent studies. The manuscript has been revised accordingly in **line 21 page**

7 to line 2 page 8 . Secondly, atomic ratio of Co, Fe, and S in CFS-ACs/CNT is also provided in Table3: Co:Fe:S= 1.31%/1.28%/19.01%.

- To investigate whether varying the content of Co, Fe, and S would change the OER catalytic activity, we prepared different contents of Co, Fe and S onto CNT, labeled to $\text{Co}_1\text{Fe}_1\text{S}_{40}$ (with 1 mmol $\text{Co}(\text{NO}_3)_2$ and 1 mmol $\text{Fe}(\text{NO}_3)_3$ as metal source), $\text{Co}_4\text{Fe}_4\text{S}_{40}$ (with 4 mmol $\text{Co}(\text{NO}_3)_2$ and 4 mmol $\text{Fe}(\text{NO}_3)_3$ as metal source), $\text{CoFe}@S_{10}$ (with 2 mmol $\text{Co}(\text{NO}_3)_2$ and 2 mmol $\text{Fe}(\text{NO}_3)_3$ as metal source and 10 mmol thioacetamide as S source); $\text{CoFe}@S_{50}$ (with 50 mmol thioacetamide as S source). The LSV curves without iR compensation of different samples was measured with a typical three-electrode system in O_2 -saturated 1.0 M KOH solutions. As shown below, altering the Co, Fe and S content can change the OER catalytic activity. Our standard samples (CFS-ACs/CNT) are prepared by a series of content control to obtain the optimal ratio of Co, Fe and S. The manuscript has been revised accordingly in line 4 -6 page 13 and line 10 -13 page 22.

Figure R20 (S19) (a) LSV curves and (b) Tafel slopes of higher amounts of Co, Fe and S onto CNT.

Q12. In Figure 4b, EXAFS data at different potentials are provided, up to 1.445 V. However, this potential corresponds to only about 20 mA/cm² over the CFS-ACs/CNT catalyst. What would happen at potentials higher than 1.445 V?

Answer: Thanks for the reviewer's valuable suggestions. We discussed that with the increase in application potential shifting from OCP, 1.245 V to 1.445 V, a slowly transformation from Fe-S coupling to Fe-O coordination environment occurs on the surface of catalyst. At the potential of 1.245 V, surface reconstruction on the surface of CoFeS_x cluster to $\text{CoFeO}(\text{OH})$ phase has happened.

In other words, from 1.245 to 1.445 V, the degree of surface reconstruction is intensified. Meanwhile, as to the potential to 1.8 V, surface reconstruction has been continuing (*Nat. Commoun.*, 2023, 14, 1949). At OCP, the coordination environment of CoFeS_x cluster is stable with Fe-S coupling. Subsequently, the catalyst surface undergoes: 1) surface S being replaced by O to form a $(\text{CoFe})\text{O}_x\text{S}_y$ composite shell; 2) as OER progresses, the sulfide surface is gradually reconstituted to form an amorphous oxide shell; 3) with the enhancement of applied potential, the degree of surface reconstitution intensifies until the catalyst is completely converted to metal oxyhydroxide species. Therefore, the evolution of this structural environment leads to changes in the Fe-S coordination environment and gradually evolves to Fe-O. EXAFS (Figure 4b) is a nice characterization technique to observe the evolution of the coordination environment: OCV (Fe-S, 1.78 Å) to 1.245 V (Fe-S(O), 1.63 Å) to 1.445 V (Fe-O(S), 1.56 Å). Therefore, at 1.6 V, even to 1.8 V, the Fe-O coordination environment maybe dominate, reflected in EXAFS data that Fe-O (S) is still slightly larger than, or even infinitely close to pure Fe-O coupling (1.50 Å). (*Nat. Catal.*, 2019, 2, 763-772; *Energy Environ. Sci.*, 2018, 11, 2945-2953; *J. Am. Chem. Soc.* 2018, 140, 11286-11292; *J. Am. Chem. Soc.* 2019, 141, 5231-5240)

Q13. I may have misunderstood the authors' intentions, but the authors proposed an OPM (or DSSM) mechanism which does not necessarily involve OOH species. However, in-situ Raman data shown in Figure 4c reveals the presence of FeOOH species at high potentials. Could the authors provide further explanations? Also, what about the presence of CoOOH species?

Answer: Thanks for the reviewer's valuable suggestions. In response to the questions raised by the reviewers, we will explain the following aspects:

Figure R21 Raman spectroscopy after OER stability of CFS-ACs/CNT.

- The *in-situ* Raman is a spectroscopic method that accurately detects the evolution of catalyst phases during electrochemical processes. Therefore, for OER process, due to the high oxidation potential environment, the catalyst surface is partially reconstituted into metal-oxygen (hydroxide) compounds (eg. FeOOH), and the generation of new chemical phases is inevitable and reasonable (*ACS Catal.*, 2022, 12, 3743-3751; *Adv. Mater.* 2023, 35, 2302462).
- Meanwhile, *in-situ* XAS (Fig. 4a,b), post-stability structural characterization (Raman (Figure S29), and XPS (Figure S30)) revealed the oxidation of the metallic Fe species to the Fe (III) phase. As shown in **Figure R21**, Raman spectroscopy after OER stability show that the main peak at 216 and 472 cm^{-1} are attributed to E_g and A_g modes of Fe-O and $\text{Co}_x\text{Fe}_y\text{-O}$ bonds, respectively (*ACS Catal.* 2022, 12, 3743–3751; *J. Electro. Soc.*, 2017, 164 (9) H621-H627). The existence of CoFeOOH phase show that the partially oxidized on the surface of CoFeS_x , which is matched with XPS spectra.

Figure R22(S31). *In-situ* Raman spectroscopy of CFS catalyst in 1.0 M KOH.

- Secondly, after a certain period of time testing, an amorphous phase will appear on the surface of the sulfide phase, forming a core-shell structure, this conclusion is also visualized in the literature with comprehensive evidence representation (*Nat. Commoun.*, 2023, 14, 1949). The

amorphous phase uses oxyhydroxide as the stable phase and is also considered to be the true active phase (*Nat. Catal.*, 2021, 4, 212-222; *Nat. Commoun.*, 2022, 13, 605). Therefore, the emergence of new species in a hydroxyl-rich environment does not conflict with the cycling mechanism of whether *OOH intermediates are involved (*Nat. Commoun.*, 2023, 14, 1949). Finally, we have revised the description of *in-situ* Raman (**Figure R22**) so that the reader can better understand the manuscript.

Q14. It is said in the manuscript that catalytic activity was enhanced due to the increased number of active sites. Is there any possibility that ‘intrinsic activity’ of the individual active sites may have been enhanced as well?

Answer: Thanks for the reviewer’s valuable suggestions. As mentioned above, CoFeS_x clusters were successfully and stably dispersed on carbon nanotubes due to the bolus effect of S atoms. Thus, a higher exposed specific surface predicts more accessible active sites. The increased number of active sites is undoubtedly advantageous for the enhancement of catalytic activity. However, in this manuscript, the emphasis is firstly on the change in spin-occupied state induced by the change in the intrinsic spin-electron number of the Co/Fe active site, exploiting the synergistic effect of the two to jointly promote O-O coupling.

Second, Co/Fe ferromagnetic elements were utilized to successfully construct a ferromagnetic catalyst and to facilitate the parallel arrangement of spin electrons for O-O coupling, thus favoring the release of the trilinear state O₂. Regarding the ‘intrinsic activity’ mentioned by the reviewer, our understanding is that the individual active sites themselves change in valence and active phase during the OER process, but the overall spin state does not change much. The enhancement is mainly reflected in the increase in OER catalytic activity induced by the evolution of the active mechanism due to a series of complex quantum spin interactions induced by the individual active sites. Therefore, we think that the above two points fully utilize the intrinsic properties of individual active sites to synergistically promote the OER reaction.

Q15. Catalyst synthesis method shown in Figure 1a might be misleading. It seems like Co, Fe and S precursors were introduced altogether at the same time, like a one-pot synthesis.

Answer: Thanks for your reasonable suggestions. We have modified the synthesis schematic in **Fig. 1a** to avoid this misinterpretation, as shown below (**Figure R23**):

Figure R23 (1a). Synthesis and morphological characterizations of CFS-ACs/CNT.

Q16. There are some trivial grammar errors and typo throughout the manuscript.

Answer: Thanks for your reasonable suggestions. We have revised the language errors in the manuscript to make it as readable as possible.

To Reviewer #3:

The paper reports synthesis of a well performing electrocatalyst that is composed of single domain ferromagnetic nanoclusters catalyst on carbon nanotube. Authors claim observing an unconventional, dual-site synergistic mechanism involving Co and Fe cations, which breaks the scaling relation that limits the performance of catalysts for oxygen evolution reaction (OER). The finding itself is definitely worth publication. However, provided discussion of the measured and computed data is sort of incoherent with plenty of results and scenarios discussed, without clear highlights. The claimed to be observed middle spin state configuration of Fe is rather unusual and would require comparison with other compounds showing similar spin state. So to say, extraordinary claims require very detailed and in-depth analysis and discussion, which unfortunately is not provided. Last, but not least, the applied simple DFT computational method is also questionable when applied to transition metal elements, especially for analysis of subtle spin arrangements.

We appreciate you for giving us such a valuable opportunity to improve on the quality of our manuscript. Thank you for your careful review of the present work and the valuable comments on our manuscript. A list of point-to-point response was prepared as bellow. We sincerely hope that the revised manuscript can meet the requirement of your suggestions.

Detailed comments:

Q1. Fe³⁺ MS should be discussed against similar spin configuration already observed in other compounds. It is not clear how such a configuration forms and if at all is permitted.

Answer: Thanks for the reviewer's valuable suggestions. We will explain and discuss the following points:

- About spin state, for 3d-metal, especially Co- and Fe-based transition metal. The spin state of active centers is related to the splitting energy (Δ_0) and the electron pairing energy (P). When the Δ_0 is higher than the P, electrons tend to pair at low-energy orbitals to form low-spin complexes. Instead, when P is higher than Δ_0 , electrons tend to transition from low-energy orbitals to high-energy orbitals, forming high-spin complexes. Since the Δ_0 and P of Co and Fe ions are relatively close, different spin state could be obtained through reasonable regulating

strategies (Figures R1a–g). Therefore, it is desired to regulate the spin state of active centers for promoting catalytic performance. However, well-known, the medium spin state, or intermediate spin state with moderate split energy, is defined as the intermediate state (number of unpaired electrons (n) between 1 and 5 between the low spin ($n \leq 1$) and high spin states ($n_{\max} = 4$ or 5). With the development of e_g orbital spin occupation states, consider $e_g=0$ as low spin, $e_g=1$ as medium spin, and $e_g=2$ as high spin. These definitions are applicable between Co, Fe-based metallic materials. (*Angew. Chem. Int. Ed.*, 2023, 62, e202216837; *J. Am. Chem. Soc.*, 2022, 144, 8204-8213; *Nat. Commun.*, 2022, 13, 605; *Precision Chemistry*, 2023, 1(7), 395-417).

Figure R1 (*Angew. Chem. Int. Ed.* 2023, 62, e202216837; *Adv. Mater.* 2020, 32, 2003297). A) The crystal field splitting of d orbitals in octahedron structure. The red and blue spheres represent O and

$3d$ metal atoms, respectively. Electronic arrangement of Co^{3+} (B–D) and Fe^{3+} (E–G) in different orbitals under low spin (B), intermediate spin (C), and high spin (D) state.

Figure R2 (A–B) Fe, Mn/N–C (M.S) (*Nat. Commun.*, 2022, 13, 605). (C–D) PQD–Fe (L.S) and o–MQFe–10:20:5 (M.S) (*Angew. Chem. Int. Ed.* 2022, 61, e202117617).

- As for Fe^{3+} M.S ($t_{2g}^4 e_g^1$), the spin-electron arrangement is similar to that in the literature (**Figure R2**), with a medium splitting energy implying that the number of paired and unpaired electrons is in equilibrium, and when the splitting energy is slightly less than the pairing energy, a pair of electrons (Figure R1E, $t_{2g}^5 e_g^0$) repels each other and splits and orbital jumps into two lone pairs of electrons with different energies (Figure R1F, $t_{2g}^4 e_g^1$). As shown in Figure R2, temperature-dependent magnetic susceptibility (M - T) curves and the ^{57}Fe Mössbauer spectroscopy are both effective techniques for detecting splitting energies and the number of unpaired electrons to evaluate the spin state. Figure R2A–B shows that the introduction of Mn regulates the transformation of the spin state of metal Fe^{3+} from L.S. to M.S. Figure R2C–D shows that the preparation of o-MQ Fe through coupling Fe-chelatednpolymer-like quantum dots (PQD-Fe) with ultrathin O-terminated MXene nanosheet ($\text{Ti}_3\text{C}_2\text{O}_x$), inducing the spin state transition of Fe^{3+} from L.S to M.S. For CoFeS_x -ACs/CNT after OER stability, with high

oxidation potential environment, Fe (II) is oxidized to Fe (III). Although Fe (II) and Fe (III) are both medium spin states, the arrangement of spin electrons is different: Fe (II) $t_{2g}^5 e_g^1$ (Figure R1C) and Fe (III) $t_{2g}^4 e_g^1$ (Figure R1F).

Q2. Introduction: scaling relation involves estimation of theoretical, thermodynamic overpotential, which is different from the measured kinetic overpotential. Please correct the discussion.

Answer: Thanks for the reviewer's valuable suggestions. We have corrected this misleading description: "Decoupling the scaling relationships of the intermediate species through the design of surface structures to activate lattice oxygen on the catalyst to follow the lattice-oxygen-mediated mechanism (LOM), is an effective strategy to drive OER activity across the adsorption energy barriers" have removed the description of dynamically-related barriers, and changed to "*Decoupling the scaling relationships between the adsorption energy of OH* and OOH* through the design of surface structures to activate lattice oxygen on the catalyst to follow the lattice-oxygen-mediated mechanism (LOM), is an effective strategy to drive OER activity*" in **line 12-15, Page 3** (*Nat. Commun.*, 2021, 12, 3992-4000; *Joule*, 2021, 5, 2164-2176).

Q3. Introduction: "catalyst itself lattice" – remove lattice. In general, there are many grammar issues, which have negative impact on overall understanding of the manuscript. Profession proofreading is advised.

Answer: Thanks for the reviewer's valuable suggestions. We apologize for the inconvenience caused by grammar issues. "catalyst itself lattice" have changed to "catalyst itself" in **line 16, Page 3**. Besides, we also checked the others grammar issues in this manuscript and revised. All modified sections are marked in yellow in this manuscript for better verification by reviewers.

Q4. Introduction, end: when discussing spins of Fe(II), LS and MS, the reason for such spin configurations should be provided.

Answer: Thanks for the reviewer's valuable suggestions. As explained in the response for the comment 1, the spin state of active centers is related to the splitting energy (Δ_0) and the electron pairing energy (P). When the Δ_0 is higher than the P, electrons tend to pair at low-energy orbitals to form low-spin complexes. Instead, when P is higher than Δ_0 , electrons tend to transition from low-energy orbitals to high-energy orbitals, forming high-spin complexes. When the splitting energy is slightly less than the pairing energy, the medium splitting energy imply the medium spin state (M.S.).

- In Introduction section: “...transforming Fe(II) from low-spin (L.S, $t_{2g}^6e_g^0$) to medium spin (M.S, $t_{2g}^5e_g^1$) through modulation of the Fe central d-orbital electron occupation...”. Fe (II) of FeS_x/CNT shows the low spin state ($t_{2g}^6e_g^0$), and Fe (II) of CFS-ACs/CNT shows the medium spin state ($t_{2g}^5e_g^1$). In Fig. 2 of Results section, temperature-dependent magnetic susceptibility (M-T) curves (**Figure R3A(Fig.2d)**) was used to evaluate the effective magnetic moment of the Fe site and to calculate the number of unpaired electrons < 1 according to the Curie-Weiss law, exhibiting a low-spin arrangement. M-T is a common and intuitive characterization technique for inferring the evolution of metal spin states. We also refer to the similar works, and our data are consistent with those in the literature (*Chem*, 2020, 6, 3440–3454; *Adv. Mater.* 2020, 32, 2003297). Similar, M-T curves was used to evaluate the number of unpaired electrons (close to 2) of CoFeS_x cluster. Meanwhile, combination with ^{57}Fe Mössbauer spectroscopy fitting of Fe (II) of CoFeS_x , predicts that the spin-electron arrangement of metallic Fe is dominated by the medium spin with the number of two unpaired electrons. Therefore, we have explained the evolution of spin states of Fe(II) from low-spin (L.S, $t_{2g}^6e_g^0$) to medium spin (M.S, $t_{2g}^5e_g^1$) in detail in the Results section.

Figure R3 (A) M-T susceptibility χ and reciprocal $1/\chi$ of CFS-ACs/CNT and FeS_x/CNT . (B) Room-temperature ^{57}Fe Mossbauer spectrum of CFS-ACs/CNT.

- The fundamental reason for the evolution of the spin state of the Fe (II) species is that the introduction of the metal Co modulates the electronic structure of the catalyst. TM orbital hybridization, valence state, and spin state could influence electronic structures of TM-based materials (*Angew. Chem. Int. Ed.* 2023, 62, e202216837). We also refer to some similar works,

and our mind are similar with those in the literature.

Figure R4 (A) $1/\chi_m$ plots of PQD-Fe (L.S.) and o-MQFe-10:20:5 (M.S.) (*Angew. Chem. Int. Ed.* 2022, 61, e202117617); ^{57}Fe Mössbauer spectra measured at 5 K of Fe-NCU (B) and Fe,Zn-NC (C) (*Nat. Catal.*, 2022, 5, 311-323); (D) Co L-edge near edge X-ray absorption fine structure (NEXAFS) spectra of CF-O, CF-SO and CF-FeSO (*Nat. Commoun.*, 2022, 13, 605); (E) $1/\chi_m$ plots and (F) ^{57}Fe Mössbauer spectra of Fe-N-C (L.S.) and Fe-N-C/Pd_{NC} (M.S.) (*Chem*, 2023, 9, 181-197).

- We think that “through modulation of the Fe central d-orbital electron occupation” summarizes spin-electron arrangement of t_{2g} (d_{xy} , d_{xz} , d_{yz}) and e_g (d_{z^2} , $d_{x^2-y^2}$) orbitals. And as mentioned above, the t_{2g} and e_g orbital electron leaps lead to a change in the electronic spin state of the metal. Therefore, we conclude that the modulation of the Fe central 3d-orbitals is indeed fundamental to the evolution of spin states.
- Following the suggestion of the reviewers, we have changed “transforming Fe (II) from low-spin (L.S, $t_{2g}^6 e_g^0$) to medium spin (M.S, $t_{2g}^5 e_g^1$) through modulation of the Fe central d-orbital electron occupation, which produces more unpaired electrons.” to “transforming Fe(II) from low-spin (L.S, $t_{2g}^6 e_g^0$) to medium spin (M.S, $t_{2g}^5 e_g^1$) through modulation of the Fe central d-orbital electron occupation, induced by the modulation of electron structure through the introduction of Co atom.” in **line 1-2, Page 6**.

Q5. Introduction, last paragraph: are these results of own studies, the reported studies. It is not clear. Anyway, this part belongs rather to conclusion part, not to the introduction.

Answer: Thanks for the reviewer's valuable suggestions. We have deleted the discussed part about this manuscript in last paragraph of Introduction section. The specific modifications are as follows: *"Inspired by the above concept, the construct of CoFeS_x clusters (1.5 ~ 3 nm sized) tightly integrated on the carbon nanotubes (CFS-ACs/CNT) as a single-domain ferromagnetic catalyst by NaBH₄ reduction followed by hydrothermal method, which might induce the transform of Fe(II) from low-spin (L.S, $t_{2g}^6 e_g^0$) to medium spin (M.S, $t_{2g}^5 e_g^1$) through modulation of the Fe central d-orbital electron occupation, induced by the modulation of electron structure through the introduction of Co atom. Benefiting to the intrinsic spin-polarized water oxidation properties of single-domain ferromagnetic catalysts, reduced kinetic energy barrier of oxygen-oxygen coupling and parallel spin electron and accelerate the release of triplet-state oxygen ($\uparrow O=O\uparrow$). Therefore, CFS-ACs/CNT catalyst will expose superior OER performance, which can balance the electrochemical activity and stability. There is no doubt that this will be a rewarding endeavor."* in line 1-10, Page 6.

Q6. Results: "The more unpaired electrons can lead to a larger degree of d-p orbital overlap between the active site electron orbital filling and the oxygen intermediate, facilitating the adsorption/desorption of oxygen intermediates of the ferromagnetic catalyst" - it is not clear how this should happen.

Answer: Thanks for the reviewer's valuable suggestions. I apologize for any misunderstanding caused by the misrepresentation. "The more unpaired electrons can lead to a larger degree of d-p orbital overlap between the active site electron orbital filling and the oxygen intermediate, facilitating the adsorption/desorption of oxygen intermediates..." have changed to *"The moderate unpaired electrons can lead to optimal d-p orbital interaction between M-3d and the O 2p, facilitating the adsorption/desorption of oxygen intermediates..."* in line 1-3, Page 11.

As shown in the **figure R5** below, we refer to the literature (*Adv. Mater.* 2020, 32, 2003297) to plot the orbital interactions between cations and the OER intermediates in order to explain the above problem more intuitively. As for the L.S ($t_{2g}^5 e_g^0$), the empty e_g orbitals lead to difficulty for the deprotonation of the oxyhydroxide group to form peroxide ions, which may be the rate-limiting. when spin state of metal site transformed to the M.S state ($t_{2g}^4 e_g^1$) with an optimal e_g orbitals occupancy of about 1.2 (*Science*, 2011, 334, 1383-1385), the catalyst exhibited optimal adsorption of key reaction intermediates and excellent OER activity. As Fe³⁺ ions transitioned from M.S state

to H.S state ($t_{2g}^3 e_g^2$), the increasement of the electrons numbers in the e_g orbital led to a weak adsorption strength of the reaction intermediates, which also made a poor OER performance (*Angew. Chem. Int. Ed.* 2023, 62, e202216837).

Figure R5 The orbital interactions between Fe^{3+} and the OER intermediates.

Q7. Results: “In which, the main peak is fitted to one doublet of D1 (blue), assigned to M.S Fe(II).”

- Please explain the fitting. Is Fe(II) MS common, ever observed? Besides, poor grammatical construction. In later text: “less than that those...” - remove “that”.

Answer: Thanks for the reviewer’s valuable suggestions. The ^{57}Fe Mössbauer spectroscopy, which is highly sensitive for probing the oxidation state, electron spin configuration (*Nat. Mater.*, 2018, 17, 625–632; *Nat. Mater.*, 2013, 12, 827–835) and coordination environment, was carried out to determine the electronic states and coordination environment of Fe in single Fe-atom catalysts. Based on the values of isomer shift (IS) and quadrupole splitting (QS), the Mössbauer spectroscopy could be well fitted. These two parameters originate from hyperfine interactions between the iron nucleus and the surrounding electronic environment and vary, therefore, with the iron oxidation state, spin state, and its chemical surrounding. (*Chemical Applications of Mössbauer Spectroscopy; Academic Press: New York and London, 1968.*) All pyrolyzed Fe-N-C catalysts comprising FeN_xC_y moieties have shown at least two distinct doublets in their Mossbauer spectra, often labeled D1 and D2 (*ACS Catal.* 2019, 9, 9359-9371). However, IS value of different fitting peaks are comparable and nondiscriminating ($0.30\text{-}0.45 \text{ mm}\cdot\text{s}^{-1}$). So, the magnitude of the quadrupole splitting (QS)

determines the evolution of the spin state. According to the Table S2 and **Figure R6**, the QS value of Fe species of CFS-ACs/CNT is 0.548, which is similar with D4 (medium-spin structure) fitting peak of Fe species of Fe, Mn-N-C (*Nat. Commun.* 2021, 12, 1734). Usually, the FeN_xC_y moieties in pyrolyzed materials are integrated in a conductive carbon matrix, which may completely change the electron density at the Fe nucleus relative to an adsorbed Fe macrocycle, even for a similar local coordination. Therefore, the difference QS ($\text{mm}\cdot\text{s}^{-1}$) values within the error range ($\pm \text{mm}\cdot\text{s}^{-1}$) are normal (*ACS Catal.* 2019, 9, 9359-9371). Therefore, according to the QS value, D1 in Fig. 2e can be assigned to medium-spin structure (Fe (II), M.S).

Figure R6. Room-temperature ^{57}Fe Mössbauer spectrum of Fe,Mn/N-C (*Nat. Commun.* 2021, 12, 1734).

Meanwhile, Fe^{2+} M.S is a common spin state of Fe and have experimentally confirmed in many previous reports. Chengzhou Zhu et al. (*Chem*, 2023, 9, 181-197) reported an unprecedented ORR catalyst consisting of Pd nanoclusters (Pd_{NC}) and Fe single atoms (Fe-N-C/PdNC). Experimental investigations and theoretical calculations indicate that the enhanced ORR activity results from the Pd_{NC} -induced spin-state transition of Fe^{2+} from low spin to intermediate spin. As shown in **Figure R7A,B**, *operando* ^{57}Fe Mössbauer spectroscopy showed that the decrease of potential indicated a drastic conversion of the spin state of Fe^{2+} in D1 from L.S to H.S. As depicted in **Figure R7C**, the ^{57}Fe Mössbauer spectrum are fitted with three doublets of D1 (green), D2 (red), and D3 (blue), assigned to L.S, M.S, and high spin (H.S) Fe^{2+} , respectively. Bin Liu et al. (*Chem*, 2020, 6, 3440-3454) have explored the exact structure of catalytic centers and provided insights into a spin-crossover-involved mechanism for oxygen reduction reaction (ORR) using *operando* Raman, X-ray absorption spectroscopies, and the developed *operando* ^{57}Fe Mössbauer spectroscopy. In addition,

there are still many excellent literature reports on the effect of Fe^{2+} M.S on the electronic structure of catalysts (Ya Cheng et al. *J. Phys.: Condens. Matter*, 2018, 30, 155403; *Nat. Chem.*, 2022, 14, 328-333; *Nat. Catal.*, 2021, 4, 10-19).

Figure R7. (a) *Operando* ^{57}Fe Mössbauer spectrum for ^{57}Fe enriched Fe-NC-S recorded at an open-circuit voltage (OCV), 0.9, 0.7, 0.5 V (versus RHE), and after ORR; (b) Content of different Fe moieties and reactive intermediates at various biases obtained from *Operando* ^{57}Fe Mössbauer spectrum. (*Chem*, 2020, 6, 3440-3454) (c) ^{57}Fe Mössbauer transmission spectra, and their deconvolution (C) of Fe-N-C and Fe-N-C/Pd_{NC}. (*Chem*, 2023, 9, 181-197)

Finally, thanks to the reviewers, we have corrected the grammar error: “...less than that those of commercial IrO_2 and FeS_x/CNT .” have changed to “less than those of commercial IrO_2 and FeS_x/CNT ” in line 9, page 13.

Q8. Results: “Consistent with previously reported⁴⁷, Fe species were oxidized to a +3 valence state...” - please explain what exactly was reported previously. Later: “is gradually shifts” – poor grammar, correct.

Answer: Thanks for the reviewer's valuable suggestions. We have corrected the above descriptive improprieties and grammatical errors in the manuscript, and yellowed the treatment:

- We have changed "Consistent with previously reported, Fe species were oxidized to a +3 valence state" to "*Consistent with previously reported, after applying the potential for a period of time, the catalyst surface reconstituted to produce amorphous Fe oxyhydroxides will be occur, where Fe species were oxidized to a +3 valence state*" in **line 7-10, page 15**.
- "a conspicuous absorption peak located at 7715 eV is gradually shifts toward..." have changed to "*a conspicuous absorption peak located at 7715 eV gradually shifts toward...*" in **line 2, page 16**.

Q9. Results: When reporting Fe-O bond lengths, please discuss which spin state these correlate the best.

Answer: We thank the reviewer's valuable suggestions. In this manuscript, we proposed that the enhancement OER activity is derived from moderate oxygen intermediate adsorption/desorption induced by the medium spin state. Despite the change in catalyst valence state before and after the OER reaction, the spin state remains unchanged and Fe³⁺-O is still stabilized in the medium spin state ($t_{2g}^4 e_g^1$). The superior OER activity was predicted to be at an e_g occupancy close to 1.2, with high covalency of transition metal-oxygen bonds (*Science, 2011, 334, 1383-1385*).

- About Fe-O bond lengths, driven by the applied voltage and rich hydroxyl molecules in solution, the sulphur atoms on the surface of CoFeS_x clusters are prone to be partially substituted by oxygen and further induce the formation of an oxygen-sulphur coexisting CoFeS(O) phase. In other words, after applying the potential for a period of time, it is inevitably form a sulphide/oxide core/shell structure. The creation of Fe-O bonds heralds the creation of an amorphous structure on the catalyst surface. Therefore, we proposed the progression from Fe-S, Fe-S(O) to Fe-O heralds the evolution of the catalyst surface structure. As for the Fe-O bond length of compounds, it is not directly related to the central point discussed in this manuscript, i.e., the evolution of the spin state, and does not directly affect the catalytic activity. Fortunately, the type of metal-oxygen covalent bonding is the one that directly affects the catalyst activity. As shown in the figure below, Yang Shao-Horn et al. reported that the peak

OER activity was predicted to be at an e_g occupancy close to 1.2, with high covalency of transition metal-oxygen bonds (*Science*, 2011, 334, 1383-1385).

Figure R8 The relation between the OER catalytic activity and the occupancy of the e_g -symmetry electron of the transition metal

Figure R9 The orbital interactions between Fe^{3+} and the OER intermediates.

- However, the transition of the spin state could affect the electronic arrangements in the e_g orbitals of transition metal ions, which affects the adsorption behavior of key intermediates on the catalyst (*Angew. Chem. Int. Ed.* 2023, 62, e202216837). The Fe (III)-O bond form between Fe active site and oxygen intermediates during OER process is then directly affecting OER activity. As shown in the figure below, we refer to the literature (*Adv. Mater.* 2020, 32, 2003297) to plot the orbital interactions between cations and the OER intermediates in order to explain the above problem more intuitively. According to the bond order law, compared with medium

spin state ($\text{Fe}^{3+} t_{2g}^4 e_g^1$), high spin state ($\text{Fe}^{3+} t_{2g}^3 e_g^2$) and low spin state ($\text{Fe}^{3+} t_{2g}^5 e_g^0$) are exposed to excessively strong oxygen intermediate adsorption, which is detrimental to the release of the result products O_2 .

Q10. “the following DFT studies were conducted comparing it to the typical $\text{Co}(\text{OH}^*)\text{-Fe}(\text{OOH}^*)$ and single-site Fe-OOH^* pathway.” - not understandable, reword. Besides, using just DFT to compute compounds with Co and Fe is not enough, especially when subtle things like spin arrangements are considered. Majority of studies apply DFT+U or hybrid functionals method.

Answer: Thanks for reviewer’s suggestion. As suggested, we carefully revised the inappropriate wording: “Furthermore, to evaluate whether Co-O-O-Fe coupling mechanism is ideal to afford the highest OER performance, DFT calculation were performed. As a comparison, typical AEM mechanisms and single-site iron sulfide models have also been considered.” has changed in **line 21-22, page 18**.

Besides, we recalculated the spin polarized DFT calculation applying DFT+U method, and conducted a careful analysis. As shown in **Figure R10a(Fig.4a)**, spin density and planar distribution maps of the spin polarization of the key oxygen intermediates ($\text{Co}(\text{O}^*)\text{-Fe}(\text{O}^*)$) are consistent with the previous ones, which still support the conclusion that the key intermediate states exhibit spin parallel arrangement.

Although the DFT+U method has a certain impact on the Gibbs free energy results, but overall it still supports the conclusion that the rate of $^*\text{O}$ to $^*\text{OOH}$ determines the energy barrier reduction of the step. Specifically, we have improved the free energy diagram of typical AEM mechanism and DSSM pathway of OER, and all calculated reaction processes showed a reasonable decrease in the free energy of the speed determination step. Not only that, for DSSM mechanism, ferromagnetic coupled $^*\text{OO}^*$ intermediates exposed lower energy barrier, which is beneficial for the release triplet-state O_2 .

Figure R10 (a) Spin density and planar distribution maps of the spin polarization of the key oxygen intermediates (Co(O*)-Fe(O*)). (b) The parallel arrangement of spin electrons in the oxygen-oxygen coupling of adjacent metal sites facilitates triplet oxygen production. (c) The free energy diagram (ΔG) of typical AEM mechanism and OPM pathway of OER including all oxygen intermediates OH*, O*, OOH* and *OO*. Insets show the spin density plot of Co(*O)-Fe(O*) over the ferromagnetic CFS model towards the transition from *O to O₂.

The corresponding calculation details of DFT+U method are as follows, which have been added to the manuscript **in line 10-13 page 25**: “To describe the strongly-localized interaction from Co-*d* and Fe-*d* electrons, the PBE-sol exchange-correlation functional was used together with an effective Hubbard parameter within the Dudarev approach with 3 eV for Fe and 3.3 eV for Co, taken from reference. (*Phys. Rev. B*, 2009, 79, 155107)”

Q11. Computations: It is not explained how different spin configurations were computed. Also authors claim computing Gibbs free energies. How then the essential entropy term was estimated?

Answer: Thank you for the thoughtful comments. The different spin configurations are obtained through magnetic and spectral characterization data. Figure 5b shows a schematic diagram of

ferromagnetic coupling and antiferromagnetic coupling of O* key intermediates for oxygen release.

We have supplemented the basic entropy calculation method and the Gibbs free energy calculation method based on the calculated hydrogen electrode in the calculation details in **line 19, page 26 to line 10, page 27**:

“The Gibbs free energy calculated by DFT is based on computational hydrogen electrode (CHE) model provided by Nørskov (*The Journal of Physical Chemistry B*, 2004, 108, 17886-17892), in which the total energy of H⁺/e⁻ is equal to $\frac{1}{2}$ H₂ at standard condition ($G_{H^+ + e^-} = \frac{1}{2} G_{H_2}$). The free energy of O₂ (g) is calculated by $G(O_2) = 2G(H_2O) - 2G(H_2) + 4.92$ eV.

In terms of each elementary step, the Gibbs free energy change is calculated via eq 1:

$$\Delta G = \Delta E + \Delta ZPE - T\Delta S \quad (1)$$

where E is the total energy obtained in DFT, ZPE is the zero-point energy, T is the temperature (298.15K), and S is the entropy. In detail, the ZPE of the adsorbate is calculated by eq 2:

$$ZPE = \frac{1}{2} \sum_i h\nu_i \quad (2)$$

the contribution from entropy TS is calculated by eq 3 with a Harmonic oscillator approximation as:

$$-TS = k_B T \sum_i \ln \left(1 - \exp\left(-\frac{h\nu_i}{k_B T}\right) \right) - \sum_i \frac{h\nu_i}{\exp\left(-\frac{h\nu_i}{k_B T}\right) - 1} \quad (3)$$

where h , and ν are Plank constant and vibrational frequency respectively.”

Q12. Overall, could be a nice paper, but if discussion is focused, key results in-depth discussed with providing convincing arguments supported by previous studies and if adequate computational approach were applied.

Answer: Thank you for your endorsement of this manuscript, and the questions you posed have been answered point-by-point above.

REVIEWER COMMENTS

Reviewer #1 (Remarks to the Author):

In the revised manuscript, I think the authors have revised all the points according to the review comments and I think it can be published without further revision.

Reviewer #2 (Remarks to the Author):

The authors responded well to all comments and requests. Therefore, I believe that it has become more convincing to the readers compared to the previous manuscript. For this reason, this manuscript is accepted for publication in Nature Communications.

Reviewer #3 (Remarks to the Author):

As I wrote last time, the paper reports synthesis of a well performing electrocatalyst that is composed of single domain ferromagnetic nanoclusters catalyst on carbon nanotube. Authors claim observing an unconventional, dual-site synergistic mechanism involving Co and Fe cations, which breaks the scaling relation that limits the performance of catalysts for oxygen evolution reaction (OER). The finding itself is definitely worth publication. However, even after revision the provided discussion of the measured and computed data is sort of incoherent with plenty of results and scenarios discussed, without clear highlights. Most of the points I raised are not addressed correctly, and I would encourage the Authors to give short and to the point answers to the concerns, instead of long text from which it is not clear what was actually changed in the revised manuscript. Among the raised points, the claimed to be observed middle spin state configuration of Fe is rather unusual and would require comparison with other compounds showing similar spin state (i.e., previous studies). So to say, extraordinary claims require very detailed and in-depth analysis and discussion, which unfortunately is not provided. At least, in the revised version Authors applied more suitable computational methodology and I am glad to see that.

Detailed comments:

- 1) Previous point: Fe³⁺ MS should be discussed against similar spin configuration already observed in other compounds. It is not clear how such a configuration forms and if at all is permitted. Please briefly discuss in the manuscript other studies showing existence of such an Fe species.
- 2) Previous point not addressed: Introduction: scaling relation involves estimation of theoretical, thermodynamic overpotential, which is different from the measured kinetic overpotential. Please correct the discussion, i.e. please mention that the theoretical estimated no not have to reflect the measured overpotentials that are driven by kinetics of the process.
- 3) In general, there are still many grammar issues, which have negative impact on overall understanding of the manuscript. Profession proofreading is highly advised!
- 4) Introduction, last paragraph: the new text is even more confusing than the old one. As I mentioned last time, Authors should briefly formulate here the scientific problem/question they attempted to tackle, and all the text reflecting their results should be shifted to the Results and Discussion section.
- 5) Previous comment: Results: "Consistent with previously reported⁴⁷, Fe species were oxidized to a +3 valence state..." - please explain what exactly was reported previously. Later: "is gradually shifts" -

poor grammar, correct. - The new text is of poor quality and not understandable.

6) Previous comment: Results: When reporting Fe-O bond lengths, please discuss which which spin state these correlate the best. This is yet another example of not providing straight and concise answer and text corrections to the comment. Simply, do the measured Fe-O bond lengths fit to the Fe MS scenario? Please address this in the text!

7) Previous comment: Computations: It is not explained how different spin configurations were computed. Also authors claim computing Gibbs free energies. How then the essential entropy term was estimated? Again, it is not addressed. The question is how different spin states were achieved in DFT calculations? Was magnetization fixed?

8) I repeat. Overall, this could be a nice paper, but if discussion is focused, text/grammar in written in reasonable and understandable way, key results are in-depth discussed with providing convincing arguments supported by previous studies and if adequate computational approach were applied. I do not see improvement here.

Reviewer #3 (Remarks to the Author):

As I wrote last time, the paper reports synthesis of a well performing electrocatalyst that is composed of single domain ferromagnetic nanoclusters catalyst on carbon nanotube. Authors claim observing an unconventional, dual-site synergistic mechanism involving Co and Fe cations, which breaks the scaling relation that limits the performance of catalysts for oxygen evolution reaction (OER). The finding itself is definitely worth publication. However, even after revision the provided discussion of the measured and computed data is sort of incoherent with plenty of results and scenarios discussed, without clear highlights. Most of the points I raised are not addressed correctly, and I would encourage the Authors to give short and to the point answers to the concerns, instead of long text from which it is not clear what was actually changed in the revised manuscript. Among the raised points, the claimed to be observed middle spin state configuration of Fe is rather unusual and would require comparison with other compounds showing similar spin state (i.e., previous studies). So to say, extraordinary claims require very detailed and in-depth analysis and discussion, which unfortunately is not provided. At least, in the revised version Authors applied more suitable computational methodology and I am glad to see that.

Answer: First of all, thank you for your recognition of our work. According to reviewer's valuable suggestions, a list of point-to-point response was prepared as bellow. We sincerely hope that the revised manuscript can meet the requirement of your suggestions.

Detailed comments:

1) Previous point: Fe³⁺ MS should be discussed against similar spin configuration already observed in other compounds. It is not clear how such a configuration forms and if at all is permitted. Please briefly discuss in the manuscript other studies showing existence of such an Fe species.

Answer: Thanks for the reviewer's valuable suggestions. With the introduction and formulation of the concept of spin, in recent years, numerous investigations have been devoted to revealing the relationship between catalytic activity and the spin configuration of catalysts, among which, the ferromagnetic element for Fe is the most studied (Table 1/S5).

Table 1/S5. The summary for medium spin state during different fields.

	Samples	Valence	Spin state	Applications	Regulatory approach	Key intermediates adsorption	Ref.
1	Fe,Mn-N-C	Fe (III)	M.S. ($t_{2g}^4e_g^1$)	Oxygen reduction reaction (ORR)	Mn-N activates the FeIII sites by electronic modulation	Antibonding π -orbital of oxygen	1
2	o-MQFe	Fe (III)	M.S. ($t_{2g}^4e_g^1$)	ORR	Axial Fe-O-Ti ligand regulation	Optimize O ₂ adsorption by FeN ₃ O	2
3	FeSA-NSC	Fe (II)	M.S. ($t_{2g}^6e_g^1$)	Nitrogen reduction reaction	Incorporation of S	Facilitating e_g electrons to penetrate the antibonding π -orbital of nitrogen	3
4	Fe-N-C/ Pd _{NC}	Fe (II)	M.S. ($t_{2g}^5e_g^1$)	ORR	Pd _{NC} -induced	Activating O–O bond through the side-on overlapping	4
5	Fe-CoOOH	Co (III)	M.S. ($t_{2g}^5e_g^1$)	Oxygen evolution reaction (OER)	Orbital occupancy of e_g	Possessing slightly strong adsorption energy	5

According to Table 1, evidently, incorporation of the second atom is an effective strategy to modulate the spin state of Fe for the optimal adsorption of oxygenated intermediates.

Therefore, we briefly discussed and analyzed the above table 1 in our revised manuscript in **line 18-23 Page 17 to 1-6 Page 18 of Results and Discussion section**, which focused on the effect of

the evolution of the spin state of Fe species on the key intermediates adsorption.

2) Previous point not addressed: Introduction: scaling relation involves estimation of theoretical, thermodynamic overpotential, which is different from the measured kinetic overpotential. Please correct the discussion, i.e. please mention that the theoretical estimated no not have to reflect the measured overpotentials that are driven by kinetics of the process.

Answer: Thanks for the reviewer's valuable suggestions. We have corrected the previous misdescription with the following changes:

- We have changed “scaling constraint of reaction intermediates OH* and OOH*..., resulting in a large theoretical overpotential of 0.37 V” to “*there is an inherent linear scaling relation (LSR) between the adsorption energy of reaction intermediates *OH and *OOH, thus rendering a minimum theoretical overpotential as high as ~0.4 eV even for the best possible material.*” (*Chem. Sci.*, 2013, 4, 2710 – 2723; *Nat. Commoun.*, 2021, 12, 3992) in **line 9-12 Page 3**.
- “Decoupling the scaling relationships between the adsorption energy of OH* and OOH* through the design of surface structures to activate lattice oxygen on the catalyst to follow the lattice-oxygen-mediated mechanism (LOM), is an effective strategy to drive OER activity” have changed to “*Compared with conventional AEM, the lattice-oxygen-mediated mechanism (LOM) breaks LSR constraint through triggering O–O coupling, which can decrease energy barrier to accelerate the OER.*” in **line 13-16 Page 3**.

3) In general, there are still many grammar issues, which have negative impact on overall understanding of the manuscript. Profession proofreading is highly advised!

Answer: We apologize for the grammatical issues that caused a barrier to understanding this work!

We've professionally proofread the entire article language at this site:

English Language Editing | Author services from Springer Nature

https://authorservices.springernature.com/language-editing/?_ga=2.212061597.822737402.1698580101-1867885648.1696667502

This document certifies that the manuscript

Dual-site segmentally synergistically catalysis mechanism: boosting CoFeS_x nanocluster for sustainable water oxidation

prepared by the authors

Siran Xu,^{1#} Sihua Feng,^{4#} Yue Yu,¹ Dongping Xue,¹ Mengli Liu,¹ Chao Wang,⁴ Kaiyue Zhao,² Bingjun Xu,² Jia-Nan Zhang^{1, 3*}

was edited for proper English language, grammar, punctuation, spelling, and overall style by one or more of the highly qualified native English speaking editors at SNAS.

This certificate was issued on **November 2, 2023** and may be verified on the SNAS website using the verification code **C8C3-A89B-E4B5-BA60-CB72**.

Neither the research content nor the authors' intentions were altered in any way during the editing process. Documents receiving this certification should be English-ready for publication; however, the author has the ability to accept or reject our suggestions and changes. To verify the final SNAS edited version, please visit our verification page at secure.authorservices.springernature.com/certificate/verify. If you have any questions or concerns about this edited document, please contact SNAS at support@as.springernature.com.

SNAS provides a range of editing, translation, and manuscript services for researchers and publishers around the world. For more information about our company, services, and partner discounts, please visit authorservices.springernature.com.

and we hope the corrections (blue highlighted) will work to your satisfaction and make it better for readers to read and share!

4) Introduction, last paragraph: the new text is even more confusing than the old one. As I mentioned last time, Authors should briefly formulate here the scientific problem/question they attempted to tackle, and all the text reflecting their results should be shifted to the Results and Discussion section.

Answer: Thanks for the reviewer's valuable suggestions. We formulate in last paragraph of *Introduction* section the scientific problem that we attempted to tackle, which is how to balance catalytic high activity and long-term stability during OER process. The specific modifications are as follows: *"In order to analyze the O-O coupling mechanism that can overcome the LSR and balance the high activity and strong stability, we chose ferromagnetic Co-Fe dual-site CoFeS_x clusters supported on carbon nanotube CNT material as a platform to catalyze OER and elucidate the intrinsic relationship between the OER activity and the spin state of each metal site.*

Meanwhile, we suspected that focusing on the preferential adsorption of key oxygen intermediates on metals is a promising research perspective for deepen understanding direct O-O coupling. Evolution of Fe(II) and Fe(III) spin states during the OER process is also considered and discussed.” and yellow highlighted in line 1-8 Page 6.

5) Previous comment: Results: “Consistent with previously reported⁴⁷, Fe species were oxidized to a +3 valence state...” - please explain what exactly was reported previously. Later: “is gradually shifts” – poor grammar, correct. - The new text is of poor quality and not understandable.

Answer: Thank you for giving us the opportunity to rework this manuscript.

➤ **Q5-1** Results: “Consistent with previously reported⁴⁷, Fe species were oxidized to a +3 valence state...” - please explain what exactly was reported previously.

Answer: Taking into account the lack of clarity here, we have changed “Consistent with previously reported⁴⁷, Fe species were oxidized to a +3 valence state...” to “*Consistent with previous reports (Hyung-Suk Oh et al. Electrode reconstruction strategy for oxygen evolution reaction: maintaining Fe-CoOOH phase with intermediate-spin state during electrolysis. Nat. Commun., 2022, 13, 605-615), oxidation state of Fe species were susceptible to changes during OER. Therefore, XPS and Raman spectra were accomplished to analyze the structure or species evolution after OER process.....*” and highlighted in line 2-4 Page 15.

➤ **Q5-2** Later: “is gradually shifts” – poor grammar, correct. - The new text is of poor quality and not understandable.

Answer: We have changed “is gradually shifts” to “*reveals that a conspicuous absorption peak located at 7715 eV gradually shifts toward high-energy with the promotion of the applied potential*” in revised manuscript. Regarding the article you mentioned is poor quality and not understandable, based on your recommendations, we have had the entire text professionally proofread for grammar at a professional organization.

6) Previous comment: Results: When reporting Fe-O bond lengths, please discuss which which spin state these correlate the best. This is yet another example of not providing straight and concise answer and text corrections to the comment. Simply, do the measured Fe-O bond lengths fit to the Fe MS scenario? Please address this in the text!

Answer: Thanks for the reviewer's suggestions. First, we need to explain that there is currently no direct evidence of correlation between the evolution of Fe-S(O) bond lengths (1.7 Å (at OCP)-1.55 Å (at 1.45 V)) during the OER process and the spin configuration of Fe species. In this manuscript, when the transition from Fe-S bond to Fe-S(O) bond, the bond lengths decrease where the electron structure of Fe species changes from Fe (II) M.S ($t_{2g}^5e_g^1$) to Fe (III) M.S ($t_{2g}^4e_g^1$). In other words, when the measured Fe-S(O) bond length is changed, the spin state of the Fe species still maintains the M.S. The detailed analysis is as follows:

- Fe-O bond, or what should reasonably be called the Fe-S(O) bond. As the OER reaction proceeds, the enhanced degree of surface remodeling induces an increase in the ratio of Fe-O/Fe-S bonds, which maps to *in-situ* XAS manifested in the shortening of bond lengths. The Fe-O bonds mentioned in this manuscript belong to the iron-oxygenhydroxides (product from surface reconstruction). The Fe-O bond lengths resulting from the combination of Fe with oxygen intermediates are maintained at 1.9 Å - 2.1 Å and are obtained by DFT calculations. The Fe-O bond lengths is a direct reflection of the strength of adsorption of the oxygen intermediates, which is closely associated with the metal spin configurations (*Adv. Mater.* 2020, 32, 2003297).
- Whereas the spin is an intrinsic electronic endowment. Currently, there is no intuitive characterization to demonstrate the change of the spin state during the OER process.
- In conclusion, even though the surface reconstruction occurs to produce metal oxygenhydroxides and thus shortening of the Fe-S(O) bond, the M.S state of Fe species is maintained throughout the catalytic process. We have also added the description in **line 20-23 Page 17**.

7) Previous comment: Computations: It is not explained how different spin configurations were computed. Also authors claim computing Gibbs free energies. How then the essential entropy term was estimated? Again, it is not addressed. The question is how different spin states were achieved in DFT calculations? Was magnetization fixed?

Q7-1: It is not explained how different spin configurations were computed. The question is how different spin states were achieved in DFT calculations? Was magnetization fixed?

Answer: Thanks for the reviewer's suggestions. **First, we need to declare that all DFT**

calculations were performed using the Vienna *Ab initio* Simulation Package (VASP). To assess the effect of the surface spin states in the DFT calculations, we introduced spin (ground state spin moment) polarized, namely, Co and Fe surfaces with spin moments. According to Jens K. Nørskov* (*ACS Catal.* 2023, 13, 3456–3462), density functional theory calculations that controlling the spin state of the surface of magnetic metals has a substantial effect on their chemical properties. Therefore, to further determine the effect of the surface spin state on the adsorbate adsorption energy, two Co-Fe surfaces with distinct spin moments of spin (ground state spin moment) and non-spin polarized were introduced for comparison. Here, non-spin polarized results have been obtained by setting the spin to zero on all Co/Fe atoms in the system and recalculating the lattice constant. As shown in Figure R1/S39, for a range of adsorbates, the adsorption energy is shown to be stronger on non-spin polarized surfaces than on spin-polarized ground state surfaces, especially in RDS step. Therefore, in this manuscript, DFT calculations with spin-polarized was introduced through the much-validated VASP for classical ground state self-consistent density functional calculations, which converge to the ground state magnetic moment. We have also added the description in **line 14-21 Page 20 and DFT calculation details.**

Figure R1/S39 Spin effect on the adsorption energy.

Second, we examined the *d*-electron occupation of the Fe site, as well as the *d*-orbital magnetic moment, which are 6.324 as well as 1.731, respectively, which coincide well with the spin configuration of Fe²⁺ (*t*_{2g}⁵*e*_g¹), and thus we think that the results of the DFT calculations can well reflect the Fe II M.S.

Third, XANES theoretical modeling was performed by using the FDMNES code.²², which coincides well with the measured XANES spectrum. This shows that the structure after convergence of our calculations is similar to the actual structure.

Q7-2: Also authors claim computing Gibbs free energies. How then the essential entropy term was estimated? Again, it is not addressed.

Answer: Thanks for the reviewer's suggestions. In this work, we determined the entropy of gas-phase molecules (H₂, O₂) from the CRC Handbook, and the adsorbate entropy (*OH, *O, *OOH) was calculated from vibrational frequencies according to Statistical Mechanics (*Charles T. Campbell* and Jason R. V. Sellers, The Entropies of Adsorbed Molecules, J. Am. Chem. Soc. 2012, 134, 18109–18115*):

$$S_M^v = R \left[\frac{\beta hc \tilde{\nu}}{e^{\beta hc \tilde{\nu}} - 1} - \ln(1 - e^{-\beta hc \tilde{\nu}}) \right]$$

where h, c and v ($\tilde{\nu}$) are Plank constant, the speed of light and vibrational frequency respectively. the contribution from entropy TS is calculated by eq 4 with a Harmonic oscillator approximation as:

$$-TS = k_B T \sum_i \ln(1 - \exp(-\frac{h\nu_i}{k_B T})) - \sum_i \frac{h\nu_i}{\exp(-\frac{h\nu_i}{k_B T}) - 1} \quad (4)$$

where h, and v are Plank constant and vibrational frequency respectively.

8) I repeat. Overall, this could be a nice paper, but if discussion is focused, text/grammar in written in reasonable and understandable way, key results are in-depth discussed with providing convincing arguments supported by previous studies and if adequate computational approach were applied. I do not see improvement here.

Answer: First of all, thank you for your recognition of our work. According to reviewer's valuable suggestions, we firstly in-depth discussed the evolution of spin state in manuscript and focus on the M.S state during OER process. Second, we enlisted the help of professional linguistic editors to linguistically proofread the entire manuscript for better understanding, we believe that the revised manuscript is reasonable and understandable. Third, we simulated with spin-polarized DFT calculations by applying DFT + U method, where the d-orbital magnetic moment of Fe are 6.324 as well as 1.731, respectively, which coincide well with the spin configuration of Fe²⁺ (*t_{2g}⁵e_g¹*). Last but not least, we would like to thank the reviewers for their input to help us refine the whole work!

REVIEWERS' COMMENTS

Reviewer #3 (Remarks to the Author):

As I have written last times, the paper reports synthesis of a well performing electrocatalyst that is composed of single domain ferromagnetic nanoclusters catalyst on carbon nanotube. Authors claim observing an unconventional, dual-site synergistic mechanism involving Co and Fe cations, which breaks the scaling relation that limits the performance of catalysts for oxygen evolution reaction (OER). The finding itself is definitely worth publication. However, even after two revisions it is rather apparent that the Authors are not able to satisfactorily address my main concerns and comments. This may be an affect on language and related understanding difficulty on the Authors side. The changes are superficial and I doubt these clarify the text to the desired extend. However, because reported findings seem to be of importance to the community I do not want to cause further delay in the publication.

Detailed comments:

- 1) Please correct grammar in the new addition marked by you in yellow. Writing such as "especially during magnetic materials"
- 2) The way how different spin states were computed is not clarified in the text.

Reviewer #3 (Remarks to the Author):

As I have written last times, the paper reports synthesis of a well performing electrocatalyst that is composed of single domain ferromagnetic nanoclusters catalyst on carbon nanotube. Authors claim observing an unconventional, dual-site synergistic mechanism involving Co and Fe cations, which breaks the scaling relation that limits the performance of catalysts for oxygen evolution reaction (OER). The finding itself is definitely worth publication. However, even after two revisions it is rather apparent that the Authors are not able to satisfactorily address my main concerns and comments. This may be an affect on language and related understanding difficulty on the Authors side. The changes are superficial and I doubt these clarify the text to the desired extend. However, because reported findings seem to be of importance to the community I do not want to cause further delay in the publication.

Answer: First of all, thank you for your recognition of our work, and thank you for the effort you have put into reviewing this manuscript. According to reviewer' s valuable suggestions, a list of point-to-point response was prepared as bellow. We sincerely hope that the revised manuscript can meet the requirement of your suggestions.

Detailed comments:

1) Please correct grammar in the new addition marked by you in yellow. Writing such as “especially during magnetic materials”

Answer: Thanks for the reviewer's valuable suggestions. We have corrected grammar and expression in the new addition marked in yellow. The specific modifications are as follows:

① **Line 6 Page 20:** “especially during magnetic materials, two Co-Fe surfaces with distinct spin moments of spin (ground state spin moment) and non-spin polarized (setting the spin to zero in the system and recalculating the lattice constant) were introduced for comparison” changed to “particularly concerning magnetic materials, two Co-Fe surfaces were introduced for comparison: one with a distinct spin moment (ground state spin moment) and the other non-spin polarized, achieved by setting the spin to zero in the system and recalculating the lattice constant”

② **Line 7-8 Page 22:** “Our work not only has achieved the simultaneous promotion of the

generation of O-O coupling intermediates and the release of O₂ ” changed to “Our work has not only achieved simultaneous promotion of O-O coupling intermediates generation and O₂ release”

③ **Line 8-9 Page 18:** “O* adsorption when Co³⁺ sites (L.S., $t_{2g}^6e_g^0$) have the dominant for *OH adsorption during OER process” changed to “O* adsorption, when Co³⁺ sites (L.S., $t_{2g}^6e_g^0$) demonstrated dominant *OH adsorption during OER process”

④ **Line 3 Page 6:** “for deepen understanding” changed to “for deepening understanding”

2) The way how different spin states were computed is not clarified in the text.

Answer: Thanks for the reviewer’s valuable suggestions. As stated in the previous two revisions, we need to declare that all DFT calculations were performed using the Vienna *Ab initio* Simulation Package (VASP). The classic computational method does not inherently assign different spin states to the model in advance. Based on our investigation and communication with experts in computational methods, it has been found that there is currently no precise definition of spin states (L.S, M.S, H.S) based on the VASP calculation method. Therefore, it is difficult for us to establish a direct relationship between controlling precise spin states via algorithmic regulation and OER activity in this work.

However, according to Jens K. Nørskov* (*ACS Catal.* 2023, 13, 3456–3462), density functional theory calculations that controlling the spin state of the surface of magnetic metals has a substantial effect on their chemical properties. Therefore, according to this literature reference, the method for controlling spin states depends on whether spin-polarized calculations are performed, introducing the ground-state spin moment. Fortunately, we are using this algorithm for DFT calculations. Not only that, to further determine the effect of the surface spin state on the adsorbate adsorption energy, two Co-Fe surfaces with distinct spin moments of spin (ground state spin moment) and non-spin polarized were introduced for comparison. (Figure S39, and detailed description has been added to the revised manuscript in the second revision. (**line 5-10 Page 20**))

Meanwhile, we examined the *d*-electron occupation of the Fe site, as well as the *d*-orbital magnetic moment, which are 6.324 as well as 1.731, respectively, which coincide well with the spin configuration of Fe²⁺ ($t_{2g}^5e_g^1$), and thus we think that the results of the DFT calculations can well reflect the Fe II M.S.

In conclusion, we performed DFT calculations using the Vienna *Ab initio* Simulation Package (VASP) with spin-polarized (ground state spin moment), and the *d*-orbital magnetic moment is

coincide well with the spin configuration of Fe^{2+} ($t_{2g}^5 e_g^1$). Therefore, we think that the results of the DFT calculations are proper. Meanwhile, in the revised manuscript, we emphasize that “OER process was simulated with spin-polarized DFT calculations by applying DFT + U method to study the OER activities of the CoFe dual-site with different mechanisms.” in **line 14-15 Page 19**. Meanwhile, to better illustrate the necessity of initiating spin, we also conducted the same OER process calculations on a non-spinning model, i.e., a model with zero magnetic moment, to demonstrate the necessity of considering spin states. **Line 5-10 Page 20**.

Finally, we sincerely thank you for your recognition of our manuscript. Your review comments have significantly improved the accuracy of our work. Moreover, inspired by your suggestions, we plan to develop a computational method for precisely defining spin states in our future work.